# Distributionally Robust Conditional Conformal Prediction

## Abstract

Conformal prediction (CP) constructs prediction sets with a marginal coverage guarantee of $1 - \alpha$, assuming the calibration distribution $P_{XY}$ and test distribution $Q_{XY}$ are identical. Under distribution shift, existing approaches align calibration and test conformal scores only at the marginal level, which helps preserve marginal coverage. However, ignoring their mismatched conditional score distributions can lead to poor conditional coverage at individual test inputs. In response, we introduce the conditional coverage gap (CCG) and its expectation over $Q_X$ to quantify the robustness of the conditional guarantee. To study how a distribution shift is propagated from data to conformal scores, we use the Wasserstein distance between $P_{XY}$ and $Q_{XY}$ to bound the expected CCG. This bound implies that an invertible transformation between $P_{XY}$ and $Q_{XY}$ via Wasserstein minimization can promote robust conditional coverage. Lastly, we implement the idea by Branched Normalizing Flow (BNF), a two-branch structure where the $X$-branch transports test inputs from $Q_X$ to $P_X$ to obtain prediction sets with conditional guarantee on $P_{Y|X}$, and the $Y$-branch inversely maps these sets with preserved conditional guarantee on $Q_{Y|X}$. Extensive experiments on nine datasets demonstrate that BNF consistently reduces CCG with improved coverage robustness across various confidence levels under distribution shift.

## 1 Introduction

Due to data noise and lack of prior knowledge, prediction uncertainty hinders applications of AI in various safety-critical domains. Conformal Prediction (CP) yields a set of possible targets rather than a single prediction to accommodate prediction uncertainty (Vovk et al., 2005; Shafer & Vovk, 2007). We focus on CP for **regression** (Lei et al., 2017). Given a trained model $h$, a score function $s(X, Y) = |h(X) - Y|$ computes the residuals (conformal scores) of $n$ calibration instances $\{(X_i, Y_i)\}_{i=1}^n$. Denoting $\tau$ the $\lceil (1 - \alpha)(n + 1) \rceil / n$ quantile of the conformal scores, a vanilla prediction set $C_{\mathrm{M}}(X_{n+1})$ of a test input $X_{n+1}$ contains all target values whose conformal scores are smaller than $\tau$. Let $P_{XY}$ and $Q_{XY}$ be calibration and test distributions in space $\mathcal{X} \times \mathcal{Y}$, respectively. If the data are independent and identically distributed (i.i.d.) so that $P_{XY} = Q_{XY}$, the prediction set $C_{\mathrm{M}}(X_{n+1})$ achieves the **marginal coverage guarantee** $\Pr(Y_{n+1} \in C_{\mathrm{M}}(X_{n+1})) \geq 1 - \alpha$. However, since $\tau$ does not depend on the specific test input $x$, $C_{\mathrm{M}}(X_{n+1})$ has constant size and lacks adaptiveness. To address the weakness, adaptive prediction set $C_{\mathrm{A}}(X_{n+1})$ aims at **conditional coverage guarantee** $\Pr(Y_{n+1} \in C_{\mathrm{A}}(X_{n+1}) | X_{n+1} = x) \geq 1 - \alpha, \forall x \in \mathcal{X}$, which provides more effective uncertainty quantification (Papadopoulos et al., 2011; Vovk, 2012).

In practice, a distribution shift ($P_{XY} \neq Q_{XY}$) can violate the i.i.d. assumption. For example, **multi-source domain generalization (MSDG)** considers $Q_{XY}$ as a random mixture of multiple source distributions (Krueger et al., 2021). In this scenario, ensuring coverage guarantees becomes both important and challenging. Let $P_V$ and $Q_V$ be the calibration and test conformal score distributions in space $\mathcal{V}$, respectively. The difference between the cumulative probabilities of $P_V$ and $Q_V$ at $\tau$ can measure the validity of marginal coverage. Various upper bounds of the discrepancy between $P_V$ and $Q_V$ are proposed to estimate the potential deviation from the nominal marginal coverage (Barber et al., 2023; Xu et al., 2025). Nevertheless, since these existing methods align calibration and test conformal scores only at the marginal level, they offer no insight into how the scores are conditionally distributed at individual inputs. As a result, these methods are unable to assess the conditional coverage of $C_{\mathrm{A}}(X_{n+1})$ under distribution shift(Figure 1(a)).

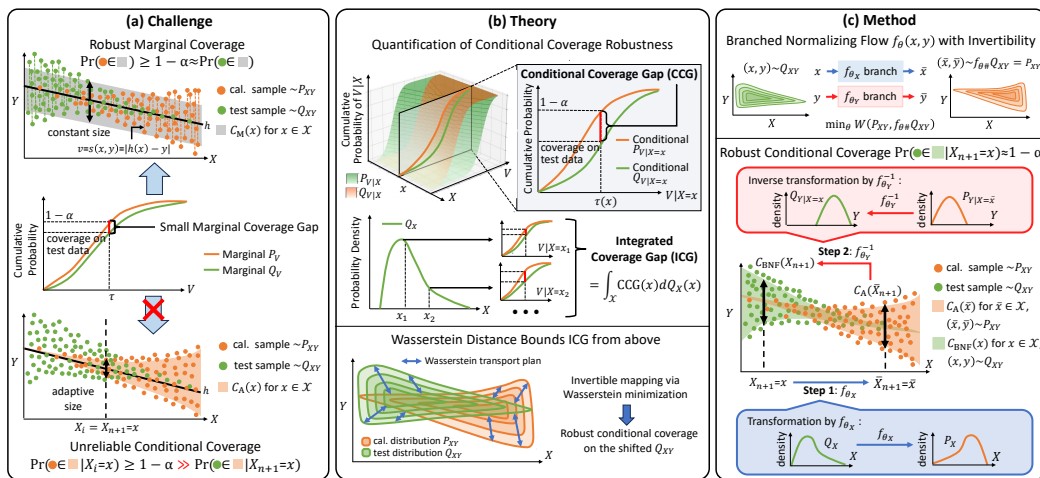

Figure 1: **(a)** Vanilla prediction set $C_M(x)$ has constant size and offers marginal coverage, which is robust if conformal score distributions $P_V$ and $Q_V$ have similar cumulative probabilities at $\tau$. Adaptive prediction set $C_A(x)$ has input-dependent size and provides conditional guarantees for calibration inputs $X_i = x$ where $i = 1, ..., n$, but may fail on non-i.i.d. test input $X_{n+1} = x$. The difference between $P_V$ and $Q_V$ can not capture the reliability of conditional coverage on the shifted test data; **(b)** Conditional coverage gap (CCG) measures $C_A(X_{n+1})$ validity at $x$ by comparing $P_{V|X=x}$ and $Q_{V|X=x}$. Integrated coverage gap (ICG) is the expectation of CCG under $Q_X$ for a holistic robustness measure. Wasserstein distance $W(P_{XY}, Q_{XY})$ bounds ICG to reveal how a distribution shift results in non-i.i.d. conformal scores. An invertible mapping between $P_{XY}$ and $Q_{XY}$ via Wasserstein minimization promotes robust conditional coverage; **(c)** Branched Normalizing Flow $f_\theta$ minimizes $W(f_{\theta\#}Q_{XY}, P_{XY})$, where $f_{\theta\#}Q_{XY}$ is a pushforward distribution. For inference, we first compute a normalized test input $\overline{X}_{n+1}$ by $f_{\theta_X}$ and generate $C_A(\overline{X}_{n+1}) \subseteq \mathcal{Y}$ with conditional guarantee on $P_{XY}$. Then, $f_{\theta_Y}^{-1}$ inversely transforms the set to $C_{BNF}(X_{n+1}) \subseteq \mathcal{Y}$ with preserved conditional coverage on $Q_{XY}$.

We aim to ensure the conditional guarantee under distribution shifts with three key contributions.

1. **Quantification of conditional coverage robustness.** We define the conditional coverage gap (CCG) in the space of conditional conformal score distributions. We further define the integrated coverage gap (ICG) as the expected CCG under the test feature distribution (Figure 1(b), 1st plot).

2. **Upper bound by Wasserstein distance.** We bound ICG with the Wasserstein distance between calibration and test distributions to reveal how a distribution shift is propagated from $\mathcal{X} \times \mathcal{Y}$ to $\mathcal{V}$. This bound implies that an invertible mapping between calibration and test distributions via Wasserstein minimization can promote robust conditional coverage (Figure 1(b) 2nd plot).

3. **Branched Normalizing Flow (BNF).** We embed the Wasserstein bound into a branched structure, defined as $f_\theta(x, y) = (f_{\theta_X}(x), f_{\theta_Y}(y)) = (\overline{x}, \overline{y})$, to transform $Q_{XY}$ to $P_{XY}$ (Figure 1(c) 1st plot). The structure does not explicitly couple the transformations of $x$ and $y$, so $f_{\theta_X}$ can compute the normalized test input without knowing the true label during inference. If the adaptive prediction set of the normalized input holds $1 - \alpha$ conditional coverage on calibration distribution, $f_{\theta_Y}^{-1}$ inversely transforms it with preserved guarantee on test distribution (Figure 1(c) 2nd plot).

To enhance the fitting ability of BNF, we propose a variant, called **Augmented BNF**, with implementation in Algorithm 1 under multi-source domain generalization (MSDG). Experiments on nine datasets cover both synthetic distribution shifts (Rana, 2013) and real-world challenges, including sales prediction across time series (Fanaee-T, 2013), traffic forecasting with mismatched data (Cui et al., 2019), medicine decision-making for different populations (Johnson et al., 2023; Pollard et al., 2018), and epidemic modeling over pandemic intervals (Deng et al., 2020). The results show that we effectively improve the robustness of the conditional guarantee.

## 2 BACKGROUND

### 2.1 ADAPTIVE CONFORMAL PREDICTION

Denote $X \in \mathcal{X} \subseteq \mathbb{R}^d$ and $Y \in \mathcal{Y} \subseteq \mathbb{R}$ the input and output random variables, respectively. With a trained regression model $h : \mathcal{X} \to \mathcal{Y}$, a score function $s : \mathcal{X} \times \mathcal{Y} \to \mathcal{V} \subseteq \mathbb{R}$ outputs conformal scores

to assess how data conform to the model $h$. We denote $V \in \mathcal{V}$ the random variable of conformal score, typically defined as the absolute residual: $V = s(X, Y) = |h(X) - Y|$. With instances $\{(X_i, Y_i)\}_{i=1}^n$ from a calibration distribution $P_{XY}$, split conformal prediction computes calibration conformal scores $V_i = s(X_i, Y_i)$ for $i = 1, ..., n$ (Papadopoulos et al., 2002). For an instance $(X_{n+1}, Y_{n+1})$ from a test distribution $Q_{XY}$, a vanilla prediction set is given by $C_{\mathrm{M}}(X_{n+1}) = \{y : s(X_{n+1}, y) \leq \tau, y \in \mathcal{Y}\}$, where $\tau$ is the $\lceil (1 - \alpha)(n + 1) \rceil / n$ quantile of $\{V_i\}_{i=1}^n$.[1] Under the i.i.d. assumption such that $P_{XY} = Q_{XY}$, $C_{\mathrm{M}}(X_{n+1})$ provides **marginal coverage guarantee** that the probability of including the true test target $Y_{n+1}$ by $C_{\mathrm{M}}(X_{n+1})$ is at least $1 - \alpha$, namely, $\Pr(Y_{n+1} \in C_{\mathrm{M}}(X_{n+1})) \geq 1 - \alpha$.

As $\tau$ is independent from test inputs, fixed-size prediction sets often underestimate uncertainty for hard samples and overestimate it for easy ones (Angelopoulos et al., 2022). Therefore, an adaptive prediction set $C_{\mathrm{A}}(X_{n+1})$ aims at improving the guarantee under the condition where $X_{n+1} = x$, $\forall x \in \mathcal{X}$. Formally, denote $\tau(x)$ the $\lceil (1 - \alpha)(n_x + 1) \rceil / n_x$ quantile of $\{V_i : X_i = x, i = 1, ...n\}$, where $n_x$ is the number of calibration samples satisfying $X_i = x$. Then, for $X_{n+1} = x$, an adaptive prediction set is given by

$$C_{\mathrm{A}}(X_{n+1}) = \{y : s(X_{n+1}, y) \leq \tau(x), y \in \mathcal{Y}\}, \tag{1}$$

and obtain **conditional coverage guarantee** (Vovk, 2012):

$$\Pr(Y_{n+1} \in C_{\mathrm{A}}(X_{n+1}) | X_{n+1} = x) \geq 1 - \alpha, \forall x \in \mathcal{X}. \tag{2}$$

However, the conditional guarantee is not practically achievable using finite calibration samples without regularity assumptions, such as Lipschitz continuity of $P_{Y|X=x}$ density (Foygel Barber et al., 2021). Hence, approximations of the conditional guarantee are extensively developed. Mondrian CP ensures $1 - \alpha$ coverage conditioned over input subspaces (Boström et al., 2021). Some methods estimate the conformal score distribution conditioned on specific test input $x$, for example, by weighting each $V_i$ based on the proximity of $X_i$ to $x$ (Lin et al., 2021; Guan, 2023; Gibbs et al., 2023). Conformal training embeds a size-based loss in the training of the model $h$ (Correia et al., 2024; Stutz et al., 2021; Bars & Humbert, 2025). Besides, advanced score functions are developed to facilitate conditional coverage in regression (Romano et al., 2019; Feldman et al., 2021). Generative models also show promise for enhancing adaptiveness, especially for multivariate output (Colombo, 2024; Fang et al., 2025; Klein et al., 2025; Thurin et al., 2025).

### 2.2 Conformal prediction for multi-source domain generalization

**Joint distribution shift** is a challenging non-i.i.d. situation where both covariate shift ($P_X \neq Q_X$) (Tibshirani et al., 2019) and concept shift ($P_{Y|X} \neq Q_{Y|X}$) (Sesia et al., 2023; Einbinder et al., 2022) can occur. Some existing works treat joint distribution shifts as perturbations on calibration data to keep marginal coverage (Gendler et al., 2021; Yan et al., 2024).

**Multi-source domain generalization (MSDG)** is a specific case of joint distribution shifts where the test distribution is in the convex hull of multiple source distributions. This scenario is extensively studied in domain adaptation theory (Zhang et al., 2019) and distributionally robust optimization (Sagawa et al., 2019). In the context of MSDG, conservative CP approaches ensure the marginal coverage under the worst-case shift (Cauchois et al., 2024; Zou & Liu, 2024). Recent work further regularizes the model $h$ for a balance between the robustness of marginal coverage and prediction efficiency (Xu et al., 2025). A related area is federated CP (Lu et al., 2023; Wen et al., 2025), where robust CP is pursued across separated sources without data centralization.

Nevertheless, how joint distribution shifts undermine the conditional coverage guarantee remains unexplored. Therefore, we focus on developing a theoretical framework to assess the robustness of conditional coverage and propose a practical solution in the presence of multiple source domains.

## 3 Theory

### 3.1 Integrated coverage gap

Under the i.i.d. assumption: $(X_{n+1}, Y_{n+1}) \sim Q_{XY} = P_{XY}$, the conditional guarantee in Eq. (2) indicates that $s(X_{n+1}, Y_{n+1}) \leq \tau(x)$ occurs with probability of at least $1 - \alpha$ when $X_{n+1} = x$:

$$\Pr(s(X_{n+1}, Y_{n+1}) \leq \tau(x) | X_{n+1} = x) \geq 1 - \alpha, \forall x \in \mathcal{X} \tag{3}$$

---

[1] Equivalently, $\tau$ can be defined as the $1 - \alpha$ quantile of $\{V_i\}_{i=1}^n \cup \{V_\infty\}$ (Vovk et al., 2005; Lei et al., 2017).

For brevity, let $P_{V|x}$ and $Q_{V|x}$ be the calibration and test conformal score distributions conditioned on an input $x$. Eq. (3) implies that $P_{V|x}$ and $Q_{V|x}$ hold the same cumulative probability at $\tau(x)$ when $P_{XY} = Q_{XY}$. Formally, denoting $F_{P_{V|x}}$ and $F_{Q_{V|x}}$ cumulative distribution functions (CDFs) of $P_{V|x}$ and $Q_{V|x}$, respectively, Eq. (3) indicates $F_{P_{V|x}}(\tau(x)) = F_{Q_{V|x}}(\tau(x)) \geq 1 - \alpha$.

To quantify how $P_{XY} \neq Q_{XY}$ impedes the conditional guarantee with $X_{n+1} = x$, we define **conditional coverage gap (CCG)** by

$$\mathrm{CCG}(P, Q, x) = |F_{P_{V|x}}(\tau(x)) - F_{Q_{V|x}}(\tau(x))|. \tag{4}$$

CCG utilizes the two CDFs, $F_{P_{V|x}}$ and $F_{Q_{V|x}}$, to assess the coverage robustness of $C_A(X_{n+1})$ when $X_{n+1} = x$. A lower CCG value indicates higher robustness. However, since test inputs are drawn from $Q_X$, evaluating CCG at a single point $x$ can not take $Q_X(x)$ at different $x$ into account. To address this, **integrated coverage gap (ICG)** is defined as the expectation of CCG under $Q_X$ by

$$\mathrm{ICG}(P, Q) = \int_{\mathcal{X}} \mathrm{CCG}(P, Q, x) \mathrm{d}Q_X(x). \tag{5}$$

By integrating CCG over $Q_X$, ICG is a comprehensive metric for coverage robustness of adaptive prediction sets. A low ICG means that conditional coverage is consistently close to $1 - \alpha$ in $\mathcal{X}$.

## 3.2 Upper bound by Wasserstein distance

We further explore how a distribution shift between $P_{XY}$ and $Q_{XY}$ in space $\mathcal{X} \times \mathcal{Y}$ is propagated to a shift between $P_{V|x}$ and $Q_{V|x}$ in space $\mathcal{V}$ for all $x \in \mathcal{X}$.

**Definition 1** (*p*-Wasserstein Distance between Population Distributions (Panaretos & Zemel, 2019))**.** *For any probability measures $\mu_X$ and $\nu_X$ defined on a metric space $(\mathcal{X}, d_{\mathcal{X}})$, where $\mathcal{X}$ is a set and $d_{\mathcal{X}}$ is a metric on $\mathcal{X}$, the Wasserstein distance of order $p \geq 1$ between $\mu_X$ and $\nu_X$ is defined by*

$$W_p(\mu_X, \nu_X) = \inf_{\gamma \in \Gamma(\mu_X, \nu_X)} \left( \int_{\mathcal{X} \times \mathcal{X}} d_{\mathcal{X}}(x_1, x_2)^p \, \mathrm{d}\gamma(x_1, x_2) \right)^{\frac{1}{p}},$$

*where $\Gamma(\mu_X, \nu_X)$ is the set of all joint probability measures $\gamma$ on $\mathcal{X} \times \mathcal{X}$ with marginals $\gamma(\mathcal{A} \times \mathcal{X}) = \mu_X(\mathcal{A})$ and $\gamma(\mathcal{X} \times \mathcal{B}) = \nu_X(\mathcal{B})$, $\forall$ measurable sets $\mathcal{A}, \mathcal{B} \subseteq \mathcal{X}$.*

The Wasserstein distance with $p = 1$ is denoted as $W$. An upper bound of the marginal coverage gap is proposed in (Xu et al., 2025). Let $L$ be the Lebesgue density bound of $P_{V|x}$ for all $x \in \mathcal{X}$ (Ross, 2011). We derive

$$\mathrm{CCG}(P, Q, x) \leq \sqrt{2L \cdot W(P_{V|x}, Q_{V|x})}. \tag{6}$$

Next, we explore how $W(P_{V|x}, Q_{V|x})$ arises from the difference in $P_{Y|x}$ and $Q_{Y|x}$ by Theorem 1.

**Theorem 1.** *Let $\mu_{XY}$ and $\nu_{XY}$ be probability measures in the metric space $(\mathcal{X} \times \mathcal{Y}, d_{\mathcal{XY}})$, where $d_{\mathcal{XY}}$ is the 2-product metric of $d_{\mathcal{X}}$ and $d_{\mathcal{Y}}$ such that $d_{\mathcal{XY}}((x_1, y_1), (x_2, y_2)) := ||(d_{\mathcal{X}}(x_1, x_2), d_{\mathcal{Y}}(y_1, y_2))||_2$. Let $s : \mathcal{X} \times \mathcal{Y} \to \mathcal{V}$ be a measurable function such that $s(x, y) = v$. In the metric space $(\mathcal{V}, d_{\mathcal{V}})$, denote $\mu_V$ the probability measure of $s(X, Y)$ for $(X, Y) \sim \mu_{XY}$. Also, let $\nu_V$ be the probability measure of $s(X, Y)$ for $(X, Y) \sim \nu_{XY}$. If $s$ has a continuity constant $\kappa$ at $x$ such that $\frac{d_{\mathcal{V}}(s(x, y_1), s(x, y_2))}{d_{\mathcal{Y}}(y_1, y_2)} \leq \kappa, \forall x \in \mathcal{X}$ and $\forall y_1, y_2 \in \mathcal{Y}$, the following inequality holds:*

$$W(\mu_{V|x}, \nu_{V|x}) \leq \kappa \cdot W(\mu_{Y|x}, \nu_{Y|x}). \tag{7}$$

A related theorem in (Xu et al., 2025) does not condition on a specific $x$. Since $\mathcal{V}, \mathcal{Y} \subseteq \mathbb{R}$, we can take the metrics $d_{\mathcal{V}}(\cdot, \cdot)$ and $d_{\mathcal{Y}}(\cdot, \cdot)$ as the absolute value of the difference. Therefore, according to Theorem 1, if the score function $s(X, Y)$ is continuous with a constant $\kappa$ such that $\frac{|s(x, y_1) - s(x, y_2)|}{|y_1 - y_2|} \leq \kappa, \forall x \in \mathcal{X}, \forall y_1, y_2 \in \mathcal{Y}$, we can derive that

$$W(P_{V|x}, Q_{V|x}) \leq \kappa \cdot W(P_{Y|x}, Q_{Y|x}). \tag{8}$$

For an intuitive explanation, a smaller $\kappa$ implies that the score function $s$ becomes less responsive to changes in $y$ conditioned on $x$. Consequently, a substantial distribution shift between $P_{Y|x}$ and $Q_{Y|x}$ will not result in a large $W(P_{V|x}, Q_{V|x})$. Combining Eq. (8) and Eq. (6), we obtain

$$\mathrm{CCG}(P, Q, x) \leq \sqrt{2\kappa L \cdot W(P_{Y|x}, Q_{Y|x})}. \tag{9}$$

Besides, as $\sqrt{W(P_{Y|x}, Q_{Y|x})} \leq W(P_{Y|x}, Q_{Y|x}) + 1/4$,[2] we can bound ICG based on Eq. (9) by

$$\text{ICG}(P, Q) \leq \sqrt{2\kappa L} \left( \int_{\mathcal{X}} W(P_{Y|x}, Q_{Y|x}) dQ_X(x) + \frac{1}{4} \right). \tag{10}$$

Eq. (10) suggests that ICG is influenced by the Wasserstein distance $W(P_{Y|x}, Q_{Y|x})$ averaged over the test input distribution $Q_X$. However, this upper bound does not fully capture the distribution shift between $P_{XY}$ and $Q_{XY}$, as it omits the effect of covariate shift, i.e., $P_X \neq Q_X$.

**Theorem 2.** *Let $\mu_{XY}$ and $\nu_{XY}$ be probability measures on the metric space $(\mathcal{X} \times \mathcal{Y}, d_{\mathcal{X}\mathcal{Y}})$. $\mu_{Y|x}$ and $\nu_{Y|x}$ are the corresponding conditional distributions of $Y$ given $X = x$. A joint distribution shift occurs between $\mu_{XY}$ and $\nu_{XY}$ such that $\mu_X \neq \nu_X$ and $\mu_{Y|X} \neq \nu_{Y|X}$. Denote $\gamma^*_{XYXY} \in \Gamma(\mu_{XY}, \nu_{XY})$ the optimal transport plan of $W(\mu_{XY}, \nu_{XY})$ and $\gamma^*_{XX}(x_1, x_2) = \int_{\mathcal{Y}^2} d\gamma^*_{XYXY}(x_1, y_1, x_2, y_2)$. If $\exists \eta > 0$ such that*

$$\int_{\mathcal{X}} W(\mu_{Y|x}, \nu_{Y|x}) d\nu_X(x) \leq \eta \int_{\mathcal{X} \times \mathcal{X}} W(\mu_{Y|x}, \nu_{Y|x}) d\gamma^*_{XX}(x, x), \tag{11}$$

*the following inequality holds that*

$$\int_{\mathcal{X}} W(\mu_{Y|x}, \nu_{Y|x}) d\nu_X(x) \leq \eta \cdot W(\mu_{XY}, \nu_{XY}). \tag{12}$$

Changing the notations $\mu$ and $\nu$ into $P$ and $Q$ in Theorem 2, we establish an upper bound for the integrated conditional Wasserstein distance in Eq. (10) as follows

$$\int_{\mathcal{X}} W(P_{Y|x}, Q_{Y|x}) dQ_X(x) \leq \eta \cdot W(P_{XY}, Q_{XY}). \tag{13}$$

Finally, combining Eq. (10) and Eq. (13), we deduce that

$$\text{ICG}(P, Q) \leq \sqrt{2\kappa L} \left( \eta \cdot W(P_{XY}, Q_{XY}) + 1/4 \right). \tag{14}$$

Eq. (14) states that ICG is bounded by $W(P_{XY}, Q_{XY})$, meaning that greater shifts in the joint distribution lead to a more significant decline in conditional coverage. However, the influence of $W(P_{XY}, Q_{XY})$ is moderated by scaling constants, which include $\kappa$, $L$, and $\eta$. The specific roles and particular implications of these constants for CP are detailed in Appendix B. The finite-sample behavior of $W(P_{XY}, Q_{XY})$ is examined in Appendix C.

## 4 METHOD

The upper bound in Eq. (14) provides a framework to ensure conditional coverage under distribution shift. Specifically, if a model $f_\theta$ transforms $Q_{XY}$ via the Wasserstein transport plan to $P_{XY}$, we have

$$(\overline{X}_{n+1}, \overline{Y}_{n+1}) := f_\theta(X_{n+1}, Y_{n+1}) \sim P_{XY}, \ \forall (X_{n+1}, Y_{n+1}) \sim Q_{XY}. \tag{15}$$

Therefore, the adaptive prediction set $C_A(\overline{X}_{n+1})$ constructed on the transformed input ensures conditional coverage with respect to $P_{XY}$. However, to achieve $1 - \alpha$ conditional coverage on $Q_{XY}$ during inference, the model $f_\theta$ must satisfy two additional requirements:

(i) $f_\theta$ can inversely transform $C_A(\overline{X}_{n+1}) \subseteq \mathcal{Y}$ with preserved conditional guarantee on $Q_{XY}$;

(ii) $f_\theta$ should not explicitly couple the transformations of $X_{n+1}$ and $Y_{n+1}$, since $Y_{n+1}$ remains unknown during inference.

### 4.1 BRANCHED NORMALIZING FLOW

Normalizing flows are widely applied techniques for invertible mapping (Kobyzev et al., 2020; Papamakarios et al., 2021). A formal definition of normalizing flows is presented in Definition 2 with a demonstration in Figure 2.

**Definition 2** (Normalizing flows (Kobyzev et al., 2020)). *Let $\mu_X$ be a probability measure in $\mathbb{R}^d$. For a measurable and invertible function $g : \mathbb{R}^d \to \mathbb{R}^d$, $\nu_X$ is the pushforward measure of $\mu_X$ through $g$, denoted as $\nu_X = g_\# \mu_X$, if $\nu_X(\mathcal{A}) = \mu_X(g^{-1}(\mathcal{A}))$ for every measurable set $\mathcal{A} \subseteq \mathbb{R}^d$. $g$ is referred to as the generative flow, and $f = g^{-1}$ is known as the normalizing flow with $\mu_X = f_\# \nu_X$.*

---

[2]When $W(P_{Y|x}, Q_{Y|x}) \geq 1$, we refine Eq. (10) into $\text{ICG}(P, Q) \leq \sqrt{2\kappa L} \int_{\mathcal{X}} W(P_{Y|x}, Q_{Y|x}) dQ_X(x)$ by $\sqrt{W(P_{Y|x}, Q_{Y|x})} \leq W(P_{Y|x}, Q_{Y|x})$. This tightens Eq. (14) to $\text{ICG}(P, Q) \leq \sqrt{2\kappa L} \cdot \eta \cdot W(P_{XY}, Q_{XY})$.

To make $f_\theta$ meet the two requirements (i) and (ii), we introduce a special normalizing flow, called **Branched Normalizing Flow (BNF)**. For a sample $(x, y)$, BNF transforms it with a branched structure such that $f_\theta(x, y) = (f_{\theta_X}(x), f_{\theta_Y}(y)) = (\overline{x}, \overline{y})$. The invertibility

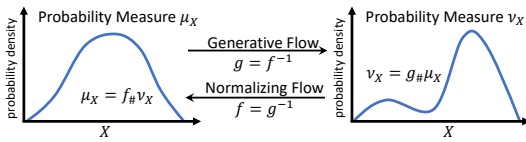

Figure 2: Invertible generative and normalizing flows.

of BNF allows that $f_\theta^{-1}(\overline{x}, \overline{y}) = (f_{\theta_X}^{-1}(\overline{x}), f_{\theta_Y}^{-1}(\overline{y})) = (x, y)$, enabling the inverse transformation of $C_A(\overline{X}_{n+1})$ and satisfying requirement (i). Besides, the parameters $\theta_X$ and $\theta_Y$ are not shared between branches, so BNF does not explicitly couple the mappings of $x$ and $y$. Therefore, the normalized test input $\overline{X}_{n+1}$ can be obtained without knowing $Y_{n+1}$ during inference, fulfilling requirement (ii).

Consider a BNF that achieves $f_{\theta\#}Q_{XY} = P_{XY}$ so that $f_{\theta_X\#}Q_X = P_X$ and $f_{\theta_Y\#}Q_{Y|x} = P_{Y|\overline{x}}$ by

$$\min_\theta W(P_{XY}, f_{\theta\#}Q_{XY}). \tag{16}$$

Then, given a test input $X_{n+1} = x$, we normalize it as $\overline{X}_{n+1} = f_{\theta_X}(x) = \overline{x}$. Since $f_{\theta\#}Q_{XY} = P_{XY}$, the transformed true target $\overline{Y}_{n+1} = f_{\theta_Y}(Y_{n+1})$, together with $\overline{X}_{n+1}$, should follow the calibration distribution, i.e, $(\overline{X}_{n+1}, \overline{Y}_{n+1}) \sim P_{XY}$, as shown in Figure 1(c) 1st plot. Therefore, the adaptive prediction set of $\overline{X}_{n+1}$ has the conditional guarantee:

$$\Pr\left(\overline{Y}_{n+1} \in C_A(\overline{X}_{n+1}) | \overline{X}_{n+1} = \overline{x}\right) \geq 1 - \alpha. \tag{17}$$

BNF then constructs a prediction set of the original input $X_{n+1}$ by including all targets whose normalized counterparts lie in $C_A(\overline{X}_{n+1})$. Specifically, we define $C_{\text{BNF}}(X_{n+1})$ as follows:

$$C_{\text{BNF}}(X_{n+1}) = \{f_{\theta_Y}^{-1}(\overline{y}) : \overline{y} \in C_A(\overline{X}_{n+1})\}, \text{ where } \overline{X}_{n+1} = f_{\theta_X}(X_{n+1}). \tag{18}$$

Proposition 6 in Appendix D and the invertibility of the univariate function $f_{\theta_Y}$ imply that

$$\overline{Y}_{n+1} \in C_A(\overline{X}_{n+1}) \iff f_{\theta_Y}^{-1}(\overline{Y}_{n+1}) \in C_{\text{BNF}}(X_{n+1}). \tag{19}$$

Consequently, since $Y_{n+1} = f_{\theta_Y}^{-1}(\overline{Y}_{n+1})$, the conditional guarantee is inherited by $C_{\text{BNF}}(X_{n+1})$:

$$\Pr\left(Y_{n+1} \in C_{\text{BNF}}(X_{n+1}) | X_{n+1} = x\right) \geq 1 - \alpha. \tag{20}$$

Even if $f_{\theta_X}$ and $f_{\theta_Y}$ do not share parameters, Wasserstein minimization in Eq. (16) considers the dependency between them. Let $y = \phi(x)$ be the ground truth mapping function of $P_{XY}$. Since $(\overline{X}_{n+1}, \overline{Y}_{n+1}) \sim P_{XY}$, we have $\overline{Y}_{n+1} = \phi(\overline{X}_{n+1}) = \phi(f_{\theta_X}(X_{n+1}))$. Thereby, even if $f_{\theta_Y}^{-1}(\overline{Y}_{n+1})$ is not explicitly conditioned on $X_{n+1}$, it implicitly depends on $X_{n+1}$ through the composition $\phi \circ f_{\theta_X}$. Further explanation with an illustrative example is provided in Appendix E.

## 4.2 ENHANCING EXPRESSIVENESS VIA GAUSSIAN NOISE AUGMENTATION

The monotonicity of the univariate $f_{\theta_Y}$ allows the equivalence in Eq. (19), but also limits its fitting ability. As a result, it struggles to optimize Eq. (16) for complex distributions, leading to unreliable conditional coverage, as shown in Appendix F.

To address this limitation, we adopt the augmentation technique proposed in (Huang et al., 2020) to gain higher expressiveness. Specifically, given a sample $(x, y)$, an augmented transformation of $y$ is defined as $f_{\theta_Y}^{\text{aug}}(y; \varepsilon) = \overline{y}$, where $\varepsilon$ is sampled from a Gaussain distribution $\mathcal{N}(0, 1)$. Meanwhile, $f_{\theta_X}$ is unchanged. We refer to this variant as **Augmented BNF**, defined as

$$f_\theta^{\text{aug}}(x, y, \varepsilon) = (f_{\theta_X}(x), f_{\theta_Y}^{\text{aug}}(y; \varepsilon)) = (\overline{x}, \overline{y}). \tag{21}$$

We implement Augmented BNF using Real NVP (Dinh et al., 2016; Huang et al., 2020), a representative coupling flow, with architectural details provided in Appendix H. Although $f_{\theta_Y}^{\text{aug}}(y; \varepsilon) = \overline{y}$ remains invertible, it does not build a monotonic relationship between $y$ and $\overline{y}$. As a result, we can not rely on Eq. (19) to preserve the conditional guarantee. To address this issue, we propose an alternative approach to obtain a prediction set for test input $X_{n+1}$ with a sampled noise $\varepsilon_{n+1}$ by defining

$$C_{\text{BNF}}^{\text{aug}}(X_{n+1}) = \left\{y : f_{\theta_Y}^{\text{aug}}(y; \varepsilon_{n+1}) \in C_A(\overline{X}_{n+1})\right\}, \text{ where } \overline{X}_{n+1} = f_{\theta_X}(X_{n+1}). \tag{22}$$

Proposition 7 in Appendix D implies that $\overline{Y}_{n+1} \in C_A(\overline{X}_{n+1}) \iff Y_{n+1} \in C_{\text{BNF}}^{\text{aug}}(X_{n+1})$. Hence, based on Eq. (17), we conclude that

$$\Pr\left(Y_{n+1} \in C_{\text{BNF}}^{\text{aug}}(X_{n+1}) | X_{n+1} = x\right) \geq 1 - \alpha. \tag{23}$$

Conditioning the $Y$ transformation on features, denoted by $f_{\theta_Y}^{\text{fea}}(y; x) = \overline{y}$, theoretically enhances expressiveness and inter-branch dependency as well. However, in practice, it exacerbates the curse of dimensionality, increasing the risk of overfitting with finite samples, as shown in Appendix G.

## 5 APPLICATION TO MULTI-SOURCE DOMAIN GENERALIZATION

In this work, we study joint distribution shift in multi-source domain generalization (MSDG) (Sagawa et al., 2019), a widely explored setting in CP (Cauchois et al., 2024; Zou & Liu, 2024; Xu et al., 2025). In MSDG, the test distribution is a random mixture within the convex hull of the source distributions. Formally, given $K$ source distributions $D_{XY}^k$ for $k = 1, .., K$, we assume the test distribution satisfies

$$Q_{XY} \in \left\{ \sum_{k=1}^{K} \lambda_k D_{XY}^k : \lambda_1, ..., \lambda_K \geq 0, \sum_{k=1}^{K} \lambda_k = 1 \right\}. \quad (24)$$

**Theorem 3.** *Let $\{\nu_{XY}^k\}_{k=1}^K$ be probability measures defined on the metric space $(\mathcal{X} \times \mathcal{Y}, d_{\mathcal{XY}})$, and let $\nu_{XY}$ lie in the convex hull of these measures, i.e., $\nu_{XY} = \sum_{k=1}^{K} \lambda_k \nu_{XY}^k$ with $\lambda_k \geq 0$ and $\sum_{k=1}^{K} \lambda_k = 1$. For any probability measure $\mu_{XY}$ on $(\mathcal{X} \times \mathcal{Y}, d_{\mathcal{XY}})$, the following inequality holds:*

$$W(\mu_{XY}, \nu_{XY}) \leq \sum_{k=1}^{K} \lambda_k W(\mu_{XY}, \nu_{XY}^k). \quad (25)$$

As outlined in (Cauchois et al., 2024; Xu et al., 2025), achieving coverage guarantee for each source distribution ensures that the coverage on the test distribution is preserved. Inspired by the principle, Theorem 3 suggests a surrogate objective for Augmented BNF by $\sum_{k=1}^{K} \lambda_k W(P_{XY}, f_\theta^{\text{aug}}{}_\# D_{XY}^k)$. Since the mixture weights $\{\lambda_k\}_{k=1}^K$ are unknown, we minimize the expectation assuming they are uniformly distributed over the simplex:

$$\min_\theta 1/K \cdot \sum_{k=1}^{K} W(P_{XY}, f_\theta^{\text{aug}}{}_\# D_{XY}^k). \quad (26)$$

We typically work with finite samples in practice. Let $\mathcal{S}_{D^k}$ be the training set from the $k$-th source distribution $D_{XY}^k$ for $k = 1, ..., n$. All training sets are of equal size. Based on the setup of split conformal prediction, all training and calibration samples are from the same distribution, so $P_{XY} = \sum_{k=1}^{K} |\mathcal{S}_{D^k}|/|\bigcup_{k=1}^{K} \mathcal{S}_{D^k}| \cdot D_{XY}^k = 1/K \cdot \sum_{k=1}^{K} D_{XY}^k$. Denote $\mathcal{S}_P$ a calibration set from $P_{XY}$. An empirical calibration distribution $\widehat{P}_{XY}$ is constructed from $\mathcal{S}_P$, as introduced in Appendix C.

During training, for each $(x, y) \in \mathcal{S}_{D^k}$, we sample a noise $\varepsilon$ from $\mathcal{N}(0, 1)$ and compute $(\bar{x}, \bar{y})$ using Eq. (21). All normalized pairs are collected in $\bar{\mathcal{S}}_{D^k}$ to construct the empirical pushforward $f_\theta^{\text{aug}}{}_\# \widehat{D}_{XY}$. Using $f_\theta^{\text{aug}}{}_\# \widehat{D}_{XY}^k$ for $k = 1, \ldots, K$ and $\widehat{P}_{XY}$, we approximate the objective in Eq. (26).

Moreover, even if the Augmented BNF perfectly achieves $f_\theta^{\text{aug}}{}_\# Q_{XY} = P_{XY}$ so that $(\bar{X}_{n+1}, \bar{Y}_{n+1}) \sim P_{XY}$, constructing a prediction set $C_A(\bar{X}_{n+1})$ that satisfies the conditional guarantee under $P_{XY}$ remains challenging with finite samples, as we introduced in Section 2.1.

In this work, we employ conformalized quantile regression (CQR) (Romano et al., 2019), which generates an adaptive prediction set $C_{\text{CQR}}(\bar{X}_{n+1})$ to approximate the $1 - \alpha$ conditional coverage specified in Eq. (17). We briefly denote the algorithm of CQR as

$$A_{\text{CQR}}\left( \bigcup_{k=1}^{K} \mathcal{S}_{D^k}, \mathcal{S}_P, \bar{X}_{n+1}, 1 - \alpha \right).$$

Crucially, the construction of $C_{\text{CQR}}(\bar{X}_{n+1})$ is independent of the Augmented BNF training and can be seamlessly integrated into our framework. Details of the CQR implementation are provided in Appendix I.

Finally, given a test set $\mathcal{S}_Q$ from $Q_{XY}$, we outline the combination of Augmented BNF + CQR in Algorithm 1.

---

**Algorithm 1** Augmented BNF + CQR under MSDG

**Require:** training sets $\mathcal{S}_{D^k}$ for $k=1, ..., K$; calibration set $\mathcal{S}_P$; test set $\mathcal{S}_Q$; $N$ epochs; $1 - \alpha$ confidence; Augmented BNF $f_\theta^{\text{aug}}$; CQR algorithm $A_{\text{CQR}}$.

---

**Training Phase:**
**for** $i=1$ to $N$ epochs **do**
    **for** $k=1$ to $K$ **do**
        Initialize $\bar{\mathcal{S}}_{D^k} \leftarrow \emptyset$
        **for** each $(x, y) \in \mathcal{S}_{D^k}$ **do**
            $(\bar{x}, \bar{y}) = f_\theta^{\text{aug}}(x, y, \varepsilon)$, where $\varepsilon \sim \mathcal{N}(0, 1)$
            $\bar{\mathcal{S}}_{D^k} \leftarrow \bar{\mathcal{S}}_{D^k} \cup \{(\bar{x}, \bar{y})\}$
        **end for**
    **end for**
    $\min_\theta \frac{1}{K} \sum_{k=1}^{K} W\left( \widehat{P}_{XY}, f_\theta^{\text{aug}}{}_\# \widehat{D}_{XY}^k \right)$
**end for**

---

**Inference Phase:**
**for** $x$ from $\mathcal{S}_Q$ **do**
    $\bar{x} = f_{\theta_X}(x)$
    $C_{\text{CQR}}(\bar{x}) = A_{\text{CQR}}\left( \bigcup_{k=1}^{K} \mathcal{S}_{D^k}, \mathcal{S}_P, \bar{x}, 1 - \alpha \right)$
    Sample $\varepsilon \sim \mathcal{N}(0, 1)$
    $C_{\text{BNF}}^{\text{aug}}(x) = \{y : f_{\theta_Y}^{\text{aug}}(y; \varepsilon) \in C_{\text{CQR}}(\bar{x})\}$
**end for**

---

# 6 EXPERIMENT

## 6.1 EXPERIMENTAL SETUP

We conduct Augmented BNF using the normflows library (Stimper et al., 2023). To estimate the empirical Wasserstein distance in Algorithm 1, we adopt the Sinkhorn algorithm (Cuturi, 2013; Knight, 2008) via the geomloss library (Feydy et al., 2019), with a brief review in Appendix J.

**Baselines.** Five methods are selected for a comprehensive comparison. Split CP (SCP) (Papadopoulos et al., 2002) guarantees marginal coverage under the i.i.d. assumption. Importance-Weighted CP (IW-CP) (Tibshirani et al., 2019) addresses covariate shifts. Worst-Case CP (WC-CP) conservatively ensures marginal coverage under joint distribution shift (Cauchois et al., 2024; Zou & Liu, 2024; Gendler et al., 2021). Wasserstein-Regularized CP (WR-CP) enhances the robustness of marginal coverage under MSDG (Xu et al., 2025). Lastly, CQR without Augmented BNF is included to demonstrate that CQR alone struggles to maintain valid conditional coverage under distribution shift. Further details on the baselines are provided in Appendix K with an illustrative example in Figure 10.

**Datasets.** We set $K = 3$ under both **synthetic** and **natural** distribution shifts. Synthetic shifts are introduced in the PTS dataset (Rana, 2013). For real-world applications, we consider (i) sales prediction over time with Bike Rental (Fanaee-T, 2013), (ii) multi-location traffic forecasting with Seattle-Loop (Cui et al., 2019), PEMSD4, and PEMSD8 (Bai et al., 2020), (iii) unbiased healthcare with MIMIC-IV (Johnson et al., 2023), eICU (Pollard et al., 2018), and data from a collaborating hospital, and (iv) epidemic modeling across pandemic phases with U.S. Influenza-like Illness (ILI) (Deng et al., 2020). Dataset details are in Appendix L.

**Evaluation metric.** The worst-slice coverage (WSC) (Cauchois et al., 2021) measures the minimal coverage over any sufficiently large slab in $\mathcal{X}$, serving as an empirical proxy for the robustness of conditional coverage, as reviewed in Appendix M. However, since WSC is evaluated only over a restricted subset of $\mathcal{X}$, it may fail to capture robustness across the entire space. Additionally, WSC captures only the minimal (i.e., most insufficient) coverage and overlooks regions where coverage may be overly conservative. To address these weaknesses, we propose **Average-slice coverage gap (ASCG)** as a practical metric to assess the robustness of conditional coverage. Since $X \in \mathcal{X} \subseteq \mathbb{R}^d$, each sample $(x, y)$ can be represented as $(x^{(1)}, x^{(2)}, \ldots, x^{(d)}, y)$. For each dimension $i \in \{1, \ldots, d\}$, we partition the test set $\mathcal{S}_Q$ into $M$ equal-sized slices along the $i$-th feature dimension. Specifically, let $\tau_{(m-1)/M}$ and $\tau_{m/M}$ denote the $(m-1)/M$ and $m/M$ quantiles of $\{x^{(i)} : (x^{(1)}, ..., x^{(d)}, y) \in \mathcal{S}_Q\}$. Then, we define the $m$-th slice in dimension $i$ as

$$\mathcal{S}_{i,m} = \{(x^{(1)}, ..., x^{(d)}, y) : x^{(i)} \in [\tau_{(m-1)/M}, \tau_{m/M}), (x^{(1)}, ..., x^{(d)}, y) \in \mathcal{S}_Q\}. \tag{27}$$

Let $c_{i,m}$ denote the number of covered samples in the slice $\mathcal{S}_{i,m}$. The ASCG is defined as

$$\text{ASCG} = \frac{1}{d} \sum_{i=1}^{d} \frac{1}{M} \sum_{m=1}^{M} \left| \frac{c_{i,m}}{|\mathcal{S}_{i,m}|} - (1 - \alpha) \right|. \tag{28}$$

If the coverage ratio $c_{i,m}/|\mathcal{S}_{i,m}|$ closely matches $1 - \alpha$ for all $i$ and $m$, then conditional coverage is approximately satisfied and ASCG remains low.

## 6.2 ROBUST CONDITIONAL COVERAGE VIA AUGMENTED BNF

We evaluate the combination of Augmented BNF and CQR, along with five baseline methods, across 10 independent trials for each dataset. The results are summarized in Figure 3, which presents box plots of coverage metrics under $1 - \alpha = 0.9$. For each trial, 100 random mixtures were generated as test sets, resulting in 1000 test performances per box. One can observe that Augmented BNF consistently achieves the lowest ASCG while maintaining marginal coverage close to 0.9, proving that Augmented BNF effectively normalizes shifted test distributions toward the calibration distribution, thereby enhancing conditional coverage robustness. We further examine the generalization ability of the Augmented BNF across varying sample sizes and $K$ values in Appendix N.

Rather than fixing $1 - \alpha = 0.9$, we explore the performance of Augmented BNF across different confidence levels. We denote $\overline{\text{ASCG}}$ the mean ASCG over 10 trials across all dataset. Figure 4 illustrates the results with $1 - \alpha$ varying from 0.1 to 0.9. The proposed method consistently achieves the most robust conditional coverage, maintaining the lowest $\overline{\text{ASCG}}$ over different confidence levels.

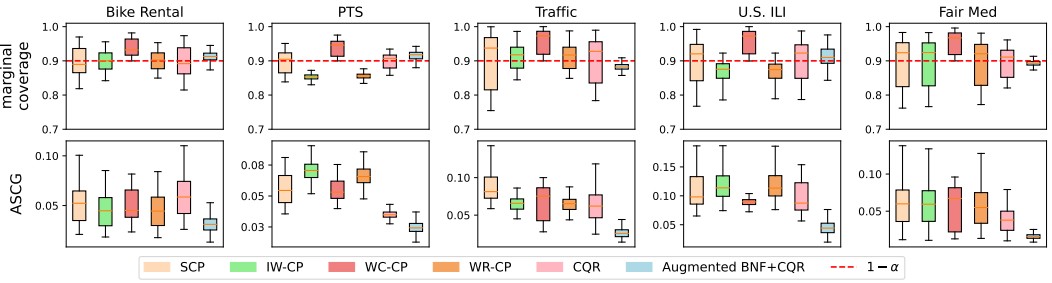

Figure 3: Marginal coverage and ASCG of Augmented BNF+CQR and five baselines with $1 - \alpha = 0.9$: the proposed method achieves the lowest ASCG and brings the marginal coverage close to the expected confidence.

Moreover, a consistent pattern emerges across all methods: $\overline{\text{ASCG}}$ tends to be higher in the mid-range of confidence levels and lower at both ends. At high confidence levels (e.g., $1 - \alpha = 0.9$), prediction intervals become wide enough to cover most possible outcomes, thereby resulting in small coverage gaps. Conversely, at low confidence levels (e.g., $1 - \alpha = 0.1$), prediction intervals are narrow and only need to capture a small subset of outcomes, making them inherently less sensitive to distribution shifts and again leading to lower coverage gaps. As a result, the coverage gap typically peaks at intermediate confidence levels, forming arch-shaped curves across the confidence spectrum from 0.1 to 0.9, as shown in Figure 4.

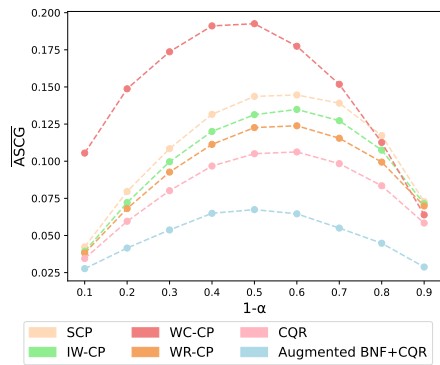

Figure 4: $\overline{\text{ASCG}}$ for $1 - \alpha$ from 0.1 to 0.9.

## 7 DISCUSSION

In practice, the loss in Eq. (26) can only be minimized empirically, and thus the transformed test distribution may not perfectly coincide with the calibration distribution. As a result, Augmented BNF cannot reduce the ASCG to zero or fully achieve the target $1 - \alpha$ marginal coverage in Figure 3. Hence, it is essential to assess how closely the attained coverage approaches $1 - \alpha$. Appendix O derives and validates marginal and conditional coverage lower bounds under imperfect transformation.

Moreover, prediction efficiency, quantified by prediction set size, is an important performance metric in CP. Reduced set size improves the likelihood of identifying the ground-truth value, while maintaining the nominal coverage level. Nevertheless, during inference, the sampled noise $\varepsilon_{n+1}$ in Eq. (22) is source-agnostic, and thus the $Y$-branch $f_{\theta_Y}^{\text{aug}}$ cannot determine the originating source distribution of a new test sample. Consequently, the prediction set $C_{\text{BNF}}^{\text{aug}}(X_{n+1})$ must expand to encompass all potential sources in order to maintain valid coverage. To address this, we propose a variant, **Augment-Conditioned BNF**, which incorporates source-specific conditioning to effectively reduce prediction set size. Appendix P provides a detailed comparison of the two kinds of BNFs.

Lastly, this work focuses on the application of our method to MSDG. Other types of distribution shift remain unexplored. We discuss this limitation, along with additional considerations, in Appendix Q.

## 8 CONCLUSION

This work proposes the Conditional Coverage Gap (CCG) to evaluate the robustness of conditional coverage at a given test input, and defines the Integrated Coverage Gap (ICG) as its expectation over the test feature distribution. We bound ICG using the Wasserstein distance $W(P_{XY}, Q_{XY})$, capturing how distribution shift propagates from the data space to the conformal score space. To ensure $1 - \alpha$ conditional coverage under shift, we introduce the Branched Normalizing Flow (BNF). The invertibility of BNF enables mapping adaptive prediction sets from $P_{XY}$ to $Q_{XY}$, while the branched structure allows input $X_{n+1}$ transformation without needing $Y_{n+1}$ at test time. BNF is applied to multi-source domain generalization (MSDG) with both synthetic and real-world distribution shifts, validating the effectiveness of our approach.

ETHICS STATEMENT

All authors have carefully read and agree to abide by the ICLR Code of Ethics. In preparing this work, we have reflected on possible ethical considerations, including issues of fairness, bias, privacy, and potential societal impacts of our methods. We have made every effort to ensure that the research was conducted responsibly and transparently, with appropriate acknowledgment of limitations and scope. We emphasize that this study does not knowingly incorporate data or methods that would compromise the rights, dignity, or safety of individuals or groups. In addition, we have considered potential risks of misuse and have aimed to present our findings in a manner that minimizes the likelihood of harmful applications.

REPRODUCIBILITY STATEMENT

We have taken deliberate steps to enhance the reproducibility of our work. The main text provides a clear description of the models, evaluation protocols, and experimental setup. Where appropriate, we have included further details in the appendix and supplementary materials to ensure that independent researchers can replicate and verify our findings. Assumptions and methodological choices are stated explicitly, and standard practices are followed to ensure comparability with prior work. Hyperparameters, evaluation criteria, and other implementation details are carefully documented to reduce ambiguity. Together, these measures are intended to support reproducibility, transparency, and scientific rigor, while allowing the community to build upon and validate our contributions.

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

# A    THE USE OF LARGE LANGUAGE MODELS

We acknowledge the use of a large language model (ChatGPT, OpenAI) for editorial assistance. Its role was limited to improving the readability of the manuscript by smoothing phrasing and correcting grammar. The research ideas, methodology, theoretical results, experiments, and technical writing were entirely conducted and authored by the researchers.

# B    INSIGHT INTO SCALING CONSTANTS OF THE ICG BOUND

We provide more intuitive explanations of the scaling constants in Eq. (14), clarifying their roles and implications for CP.

First, the term $L$, representing the Lebesgue density bound of $P_{V|x}$, captures the concentration of conformal scores at $x$. A higher $L$ indicates that the calibration scores are tightly clustered around certain values of $v$, which makes the conditional coverage more sensitive to distribution shifts in test conformal scores. Hence, this highlights how the shape of the calibration conformal score distribution directly influences coverage robustness.

Second, $\kappa$ provides an interpretation of how the score function $s(x, y)$ influences robustness under distribution shift. Specifically, the continuity constant $\kappa \geq (|s(x, y_1) - s(x, y_2)|)/(|y_1 - y_2|)$ for all $y_1, y_2 \in \mathcal{Y}, x \in \mathcal{X}$. It captures the sensitivity of the score function $s(x, y)$ to changes in the label $y$, given a fixed input $x$. A smaller $\kappa$ implies that the conformal score is relatively insensitive to variations in the label, meaning that even under a large concept shift (i.e., large $W(P_{Y|x}, Q_{Y|x})$), the induced shift in conformal scores $W(P_{V|x}, Q_{V|x})$ remains small.

Lastly, the term $\eta$, introduced in Eq. (12), quantifies the extent to which the concept shift contributes to the overall joint distribution shift. A smaller $\eta$ indicates that most of the distributional difference between $P_{XY}$ and $Q_{XY}$ does not stem from the difference between $P_{Y|x}$ and $Q_{Y|x}$ for $x \in \mathcal{X}$. In such cases, the impact of $W(P_{XY}, Q_{XY})$ on the coverage gap is limited, and accordingly, the upper bound in Eq. (14) becomes tighter.

# C    FINITE-SAMPLE APPROXIMATION OF WASSERSTEIN DISTANCE

In practice, the population forms of calibration and test distributions are typically inaccessible, so we may approximate $W(P_{XY}, Q_{XY})$ based on empirical distributions.

**Definition 3** (*p-Wasserstein Distance between Empirical Distributions (Panaretos & Zemel, 2019)*). *Let $\{x_i\}_{i=1}^n \sim \mu_X$ and $\{x'_j\}_{j=1}^m \sim \nu_X$ be i.i.d. samples from two distributions on a metric space $(\mathcal{X}, d_\mathcal{X})$. The Dirac measure $\varepsilon_x$ is the point mass at $x \in \mathcal{X}$. The empirical measures are defined as*

$$\widehat{\mu}_X = \frac{1}{n} \sum_{i=1}^n \varepsilon_{x_i}, \quad \widehat{\nu}_X = \frac{1}{m} \sum_{j=1}^m \varepsilon_{x'_i}.$$

$C \in \mathbb{R}^{n \times m}$ *is a cost matrix where each element $C_{ij} = d_\mathcal{X}(x_i, x'_j)$ measures the distance between sample $x_i$ from $\widehat{\mu}_X$ and $x'_j$ from $\widehat{\nu}_X$. Let $\gamma \in \mathbb{R}^{n \times m}$ be a transportation plan matrix, where each $\gamma_{ij} \geq 0$ represents the mass transported from $x_i$ to $x'_j$. The set of admissible transport plans is*

$$\Gamma(\widehat{\mu}_X, \widehat{\nu}_X) = \left\{ \gamma \in \mathbb{R}_{\geq 0}^{n \times m} \,\middle|\, \sum_{j=1}^m \gamma_{ij} = \frac{1}{n}, \ \sum_{i=1}^n \gamma_{ij} = \frac{1}{m} \right\}.$$

*The p-Wasserstein distance between empirical distributions $\widehat{\mu}_X$ and $\widehat{\nu}_X$ is then given by*

$$W_p(\widehat{\mu}_X, \widehat{\nu}_X) = \left( \min_{\gamma \in \Gamma(\widehat{\mu}_X, \widehat{\nu}_X)} \sum_{i=1}^n \sum_{j=1}^m \gamma_{ij} \, C_{ij}^p \right)^{1/p}.$$

Let $\widehat{P}_{XY}$ and $\widehat{Q}_{XY}$ be the empirical distributions based on $n$ and $m$ i.i.d. samples drawn from $P_{XY}$ and $Q_{XY}$, respectively. Our goal is to bound the deviation between the empirical and population

Wasserstein distances, i.e., to analyze how $W(\widehat{P}_{XY}, \widehat{Q}_{XY})$ converges to $W(P_{XY}, Q_{XY})$ as $n$ increases.

**Definition 4** (Upper Wasserstein Dimension (Dudley, 1969)). *Given a set $\mathcal{A} \subseteq \mathcal{X}$, the $\epsilon$-covering number, denoted $\mathcal{N}_\epsilon(\mathcal{A})$, is the smallest $n$ such that $n$ closed balls, $\mathcal{U}_1, ..., \mathcal{U}_n$, of diameter $\epsilon$ achieve $\mathcal{A} \subseteq \cup_{1 \leq i \leq m} \mathcal{U}_i$. For a distribution $\mu_X$ in $\mathcal{X}$, the $(\epsilon, \zeta)$-dimension is $d_\epsilon(\mu_X, \zeta) = -\log(\inf\{\mathcal{N}_\epsilon(\mathcal{A}) : \mu_X(\mathcal{A}) \geq 1 - \zeta\})/\log \epsilon$. The upper Wassersteion dimension with $p = 1$ is*

$$d_W(\mu_X) = \inf\{\varphi \in (2, \infty) : \limsup_{\epsilon \to 0} d_\epsilon(\mu_X, \epsilon^{\frac{\varphi}{\varphi - 2}}) \leq \varphi\}. \tag{29}$$

**Theorem 4.** *Given a probability measure $\mu_X$ in space $\mathcal{X}$, let $\sigma > d_W(\mu_X)$. If $\widehat{\mu}_X$ is an empirical measure corresponding to $n$ i.i.d. samples from $\mu_X$, $\exists \lambda \in \mathbb{R}$ such that $\mathbb{E}[W(\mu_X, \widehat{\mu}_X)] \leq \lambda n^{-1/\sigma}$. Furthermore, for $t > 0$, $\Pr(W(\mu_X, \widehat{\mu}_X) \geq \mathbb{E}[W(\mu_X, \widehat{\mu}_X)] + t) \leq e^{-2nt^2}$ (Weed & Bach, 2019).*

**Theorem 5.** *Given probability measures $\mu_X$ and $\nu_X$ in space $\mathcal{X}$, let $\sigma_\mu > d_W(\mu_X)$ and $\sigma_\nu > d_W(\nu_X)$. Denote $\widehat{\mu}_X$ and $\widehat{\nu}_X$ empirical measures corresponding to $n$ and $m$ i.i.d. samples from $\mu_X$ and $\nu_X$, respectively. For $t_\mu, t_\nu > 0$, $\exists \lambda_\mu, \lambda_\nu > 0$ with probability at least $(1 - e^{-2nt_\mu^2})(1 - e^{-2mt_\nu^2})$ that*

$$|W(\mu_X, \nu_X) - W(\widehat{\mu}_X, \widehat{\nu}_X)| \leq \lambda_\mu n^{-1/\sigma_\mu} + \lambda_\nu m^{-1/\sigma_\nu} + t_\mu + t_\nu. \tag{30}$$

A related theorem is proposed in (Xu et al., 2025), though without accounting for the signs of $\lambda_\mu$ and $\lambda_\nu$. Based on Theorem 5, if $\sigma_P > d_W(P_{XY})$ and $\sigma_Q > d_W(Q_{XY})$, for $t_P, t_Q > 0$, there are $\lambda_P, \lambda_Q > 0$ with a probability at least $(1 - e^{-2nt_P^2})(1 - e^{-2mt_Q^2})$ that

$$\left| W(P_{XY}, Q_{XY}) - W(\widehat{P}_{XY}, \widehat{Q}_{XY}) \right| \leq \lambda_P n^{-1/\sigma_P} + \lambda_Q m^{-1/\sigma_Q} + t_P + t_Q. \tag{31}$$

As $n$ and $m$ increase, the bound in Eq. (31) decreases, thereby improving the approximation of the empirical Wasserstein distance. At the same time, the probability $(1 - e^{-2nt_P^2})(1 - e^{-2mt_Q^2})$ increases, indicating that the bound holds with higher confidence.

# D    ADDITIONAL THEORETICAL STATEMENTS

## D.1    SUPPORTING PROPOSITIONS

**Proposition 6.** *Let $f : \mathcal{X} \to \mathcal{Y}$ be an invertible univariate function, where $\mathcal{X}, \mathcal{Y} \subseteq \mathbb{R}$. Let $C = [y_{\text{lo}}, y_{\text{hi}}] \subseteq \mathcal{Y}$ be a closed interval. Then for any $y \in \mathcal{Y}$, the following equivalence holds:*

$$y \in C \quad \Longleftrightarrow \quad f^{-1}(y) \in \{x \in \mathcal{X} : f(x) \in C\}.$$

*Proof.* Since $f$ is an invertible univariate function, it must be strictly monotonic—either strictly increasing or strictly decreasing.

**Case 1:** Suppose $f$ is strictly increasing. Then $f^{-1}$ is also strictly increasing.

$\Rightarrow$ If $y \in C = [y_{\text{lo}}, y_{\text{hi}}]$, then by monotonicity,
$$f^{-1}(y_{\text{lo}}) \leq f^{-1}(y) \leq f^{-1}(y_{\text{hi}}),$$
so $f^{-1}(y) \in [f^{-1}(y_{\text{lo}}), f^{-1}(y_{\text{hi}})]$. Since $f$ is strictly increasing, this implies
$$[f^{-1}(y_{\text{lo}}), f^{-1}(y_{\text{hi}})] = \{x \in \mathcal{X} : f(x) \in C\}$$
and thus $f^{-1}(y) \in \{x \in \mathcal{X} : f(x) \in C\}$.

$\Leftarrow$ If $f^{-1}(y) \in \{x \in \mathcal{X} : f(x) \in C\}$, then equivalently we can derive $y \in C$.

**Case 2:** Suppose $f$ is strictly decreasing. Then $f^{-1}$ is also strictly decreasing.

$\Rightarrow$ If $y \in C = [y_{\text{lo}}, y_{\text{hi}}]$, then
$$f^{-1}(y_{\text{lo}}) \geq f^{-1}(y) \geq f^{-1}(y_{\text{hi}}),$$
so $f^{-1}(y) \in [f^{-1}(y_{\text{hi}}), f^{-1}(y_{\text{lo}})]$. Again, since $f$ is decreasing,
$$[f^{-1}(y_{\text{hi}}), f^{-1}(y_{\text{lo}})] = \{x \in \mathcal{X} : f(x) \in C\},$$
which implies $f^{-1}(y) \in \{x \in \mathcal{X} : f(x) \in C\}$.

$\Leftarrow$ If $f^{-1}(y) \in \{x \in \mathcal{X} : f(x) \in C\}$, then again $y \in C$.

In either case, the equivalence holds. $\qquad \square$

**Proposition 7.** *Let $f : \mathcal{X} \to \mathcal{Y}$ be a univariate function, where $\mathcal{X}, \mathcal{Y} \subseteq \mathbb{R}$. Let $C \subseteq \mathcal{Y}$ be a closed interval. Then for $a \in \mathcal{X}$, it holds that:*

$$f(a) \in C \iff a \in \{x \in \mathcal{X} : f(x) \in C\}.$$

*Proof.* The statement is a direct consequence of the definition of the set $\{x \in \mathcal{X} : f(x) \in C\}$. By definition, $a$ belongs to this set if and only if $a \in \mathcal{X}$ and $f(a) \in C$. Since $a \in \mathcal{X}$ is already assumed, the condition reduces to: $f(a) \in C \iff a \in \{x \in \mathcal{X} : f(x) \in C\}$. $\qquad \square$

We visualize Proposition 6 and Proposition 7 in Figure 5.

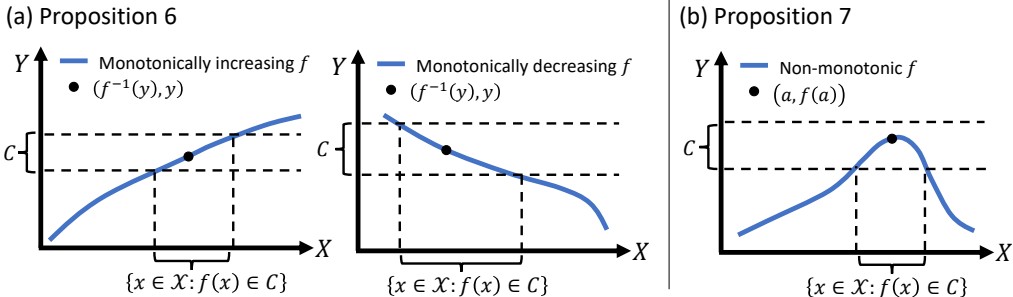

Figure 5: Characterization of preimage membership under **(a)** monotonic and **(b)** non-monotonic functions.

### D.2 PROOF OF THEOREM 1

*Proof.* Let $\mu_{XY}$ and $\nu_{XY}$ be probability measures on the metric space $(\mathcal{X} \times \mathcal{Y}, d_{\mathcal{X}\mathcal{Y}})$, where $d_{\mathcal{X}\mathcal{Y}}((x_1, y_1), (x_2, y_2)) := ||(d_{\mathcal{X}}(x_1, x_2), d_{\mathcal{Y}}(y_1, y_2))||_2$. Let $s : \mathcal{X} \times \mathcal{Y} \to \mathcal{V}$ be a measurable function such that $s(x, y) = v$. In metric space $(\mathcal{V}, d_{\mathcal{V}})$, denote $\mu_V$ the probability measure of $s(X, Y)$ for $(X, Y) \sim \mu_{XY}$. Also, let $\nu_V$ be the probability measure of $s(X, Y)$ for $(X, Y) \sim \nu_{XY}$. Denote $\Gamma_{V|x} = \Gamma(\mu_{V|x}, \nu_{V|x})$ and $\Gamma_{Y|x} = \Gamma(\mu_{Y|x}, \nu_{Y|x})$. By Theorem 1 in (Xu et al., 2025), we derive

$$W(\mu_{V|x}, \nu_{V|x}) = \inf_{\gamma \in \Gamma_{V|x}} \int_{\mathcal{V} \times \mathcal{V}} d_{\mathcal{V}}(v_1, v_2) \, \mathrm{d}\gamma(v_1, v_2)$$
$$= \inf_{\gamma \in \Gamma_{Y|x}} \int_{\mathcal{Y} \times \mathcal{Y}} d_{\mathcal{V}}(s(x, y_1), s(x, y_2)) \, \mathrm{d}\gamma(y_1, y_2). \tag{32}$$

Consider $\gamma^* \in \Gamma_{Y|x}$ is the optimal transport plan for $W(\mu_{Y|x}, \nu_{Y|x})$. However, $\gamma^*$ is not necessarily optimal for obtaining $W(\mu_{V|x}, \nu_{V|x})$ in Eq. (32), so we have

$$W(\mu_{V|x}, \nu_{V|x}) \leq \int_{\mathcal{Y} \times \mathcal{Y}} d_{\mathcal{V}}(s(x, y_1) - s(x, y_2)) \, \mathrm{d}\gamma^*(y_1, y_2). \tag{33}$$

Given that the function $s$ is continuous with constant $\kappa$ conditioned on $x$, we have $\frac{d_{\mathcal{V}}(s(x, y_1), s(x, y_2))}{d_{\mathcal{Y}}(y_1 - y_2)} \leq \kappa, \forall x \in \mathcal{X}, y_1, y_2 \in \mathcal{Y}$, so the following inequality holds that

$$\int_{\mathcal{Y} \times \mathcal{Y}} d_{\mathcal{V}}(s(x, y_1), s(x, y_2)) \, \mathrm{d}\gamma^*(y_1, y_2) \leq \int_{\mathcal{Y} \times \mathcal{Y}} \kappa \cdot d_{\mathcal{Y}}(y_1, y_2) \, \mathrm{d}\gamma^*(y_1, y_2) = \kappa \cdot W(\mu_{Y|x}, \nu_{Y|x}). \tag{34}$$

Finally, combining Eq. (33) and Eq. (34), we can conclude that

$$W(\mu_{V|x}, \nu_{V|x}) \leq \kappa \cdot W(\mu_{Y|x}, \nu_{Y|x}). \tag{35}$$

$\qquad \square$

### D.3 PROOF OF THEOREM 2

*Proof.* Let $\mu_{XY}$ and $\nu_{XY}$ be probability measures on the metric space $(\mathcal{X} \times \mathcal{Y}, d_{\mathcal{X}\mathcal{Y}})$, where $d_{\mathcal{X}\mathcal{Y}}((x_1, y_1), (x_2, y_2)) := ||(d_{\mathcal{X}}(x_1, x_2), d_{\mathcal{Y}}(y_1, y_2))||_2$. A joint distribution shift results in $\mu_X \neq \nu_X, \mu_{Y|X} \neq \nu_{Y|X}$.

For any $\gamma_{XYXY} \in \Gamma(\mu_{XY}, \nu_{XY})$, denote $\gamma_{XX}(x_1, x_2) = \int_{\mathcal{Y}^2} \mathrm{d}\gamma_{XYXY}(x_1, y_1, x_2, y_2)$. Thereby, we can derive

$$\int_{\mathcal{X}^2 \times \mathcal{Y}^2} d_{\mathcal{X}\mathcal{Y}}((x_1, y_1), (x_2, y_2))\mathrm{d}\gamma_{XYXY}(x_1, y_1, x_2, y_2)$$

$$\geq \int_{\mathcal{X}^2 \times \mathcal{Y}^2} d_{\mathcal{Y}}(y_1, y_2)\mathrm{d}\gamma_{XYXY}(x_1, y_1, x_2, y_2)$$

$$\geq \int_{\mathcal{X}^2 \times \mathcal{Y}^2} d_{\mathcal{Y}}(y_1, y_2)I(x_1 = x_2)\mathrm{d}\gamma_{XYXY}(x_1, y_1, x_2, y_2) \qquad (36)$$

$$= \int_{\mathcal{X}^2} \left( \int_{\mathcal{Y}^2} d_{\mathcal{Y}}(y_1, y_2)\mathrm{d}\gamma_{YY|x_1 x_2}(y_1, y_2) \right) I(x_1 = x_2)\mathrm{d}\gamma_{XX}(x_1, x_2)$$

$$= \int_{\mathcal{X}^2} \left( \int_{\mathcal{Y}^2} d_{\mathcal{Y}}(y_1, y_2)\mathrm{d}\gamma_{YY|x_1 x_1}(y_1, y_2) \right) \mathrm{d}\gamma_{XX}(x_1, x_1).$$

Consider $\gamma^*_{XYXY} \in \Gamma(\mu_{XY}, \nu_{XY})$ that satisfies

$$W(\mu_{XY}, \nu_{XY}) = \int_{\mathcal{X}^2 \times \mathcal{Y}^2} d_{\mathcal{X}\mathcal{Y}}((x_1, y_1), (x_2, y_2))\mathrm{d}\gamma^*_{XYXY}(x_1, y_1, x_2, y_2). \qquad (37)$$

However, $\gamma^*_{YY|x_1 x_1}$ is not necessarily the optimal transport plan of $W(\mu_{Y|x_1}, \nu_{Y|x_1}), \forall x_1 \in \mathcal{X}$, so

$$W(\mu_{Y|x_1}, \nu_{Y|x_1}) \leq \int_{\mathcal{Y}^2} d_{\mathcal{Y}}(y_1, y_2)\mathrm{d}\gamma^*_{YY|x_1 x_1}(y_1, y_2). \qquad (38)$$

Therefore, after plugging Eq. (37) and Eq. (38) into Eq. (36) and simplifying $x_1$ as $x$, we obtain

$$W(\mu_{XY}, \nu_{XY}) \geq \int_{\mathcal{X}^2} W(\mu_{Y|x}, \nu_{Y|x})\mathrm{d}\gamma^*_{XX}(x, x). \qquad (39)$$

Given $\eta > 0$ that satisfies

$$\eta \int_{\mathcal{X}^2} W(\mu_{Y|x}, \nu_{Y|x})\mathrm{d}\gamma^*_{XX}(x, x) \geq \int_{\mathcal{X}} W(\mu_{Y|x}, \nu_{Y|x})\mathrm{d}\nu_X(x), \qquad (40)$$

we can consequently prove

$$\eta \cdot W(\mu_{XY}, \nu_{XY}) \geq \int_{\mathcal{X}} W(\mu_{Y|x}, \nu_{Y|x})\mathrm{d}\nu_X(x). \qquad (41)$$

$\square$

We would like to further justify the necessity of introducing $\eta$ to satisfy Eq. (40).

Considering $\int_{\mathcal{X}}^2 \mathrm{d}\gamma^*_{XX}(x, x) = \int_{\mathcal{X}}^2 I(x_1 = x_2)\mathrm{d}\gamma^*_{XX}(x_1, x_2)$, we denote

$$\psi(\mathcal{A}) = \gamma^*_{XX} \left( \{(x_1, x_2) \in \mathcal{X}^2 : x_1 = x_2 \in \mathcal{A}\} \right) = \gamma^*_{XX}(\mathcal{A} \times \mathcal{A}), \forall \mathcal{A} \subset \mathcal{X}. \qquad (42)$$

As $\nu_X$ is a projection of $\gamma^*_{XX}$, we have $\nu_X(\mathcal{A}) = \gamma^*_{XX}(\mathcal{A} \times \mathcal{X}) \geq \psi(\mathcal{A})$. By the Radon-Nikodym theorem (Fonseca & Leoni, 2007), there exists a density $\rho(x) \geq 0$ such that

$$\psi(\mathcal{A}) = \int_{\mathcal{A}} \rho(x)\mathrm{d}\nu_X(x). \qquad (43)$$

Since $\psi(\mathcal{A}) \leq \nu(\mathcal{A})$, we can derive $\int_{\mathcal{A}} \rho(x)\mathrm{d}\nu_X(x) \leq \int_{\mathcal{A}} 1\mathrm{d}\nu_X(x)$ for all $\mathcal{A}$. This forces $\rho(x) \leq 1$ almost everywhere on $\nu_X$. As a result, we conclude that

$$\int_{\mathcal{X}^2} W(\mu_{Y|x}, \nu_{Y|x})\mathrm{d}\psi(x) = \int_{\mathcal{X}} W(\mu_{Y|x}, \nu_{Y|x})\rho(x)\mathrm{d}\nu_X(x) \leq \int_{\mathcal{X}} W(\mu_{Y|x}, \nu_{Y|x})\mathrm{d}\nu_X(x). \qquad (44)$$

Therefore, we introduce a constant $\eta$ to reverse the inequality in Eq. (44).

## D.4    PROOF OF THEOREM 3

*Proof.* For each $k \in \{1, ..., K\}$, denote $\gamma^k \in \Gamma(\mu_{XY}, \nu_{XY}^k)$ the optimal transport plan realizing $W(\mu_{XY}, \nu_{XY}^k)$ such that

$$W(\mu_{XY}, \nu_{XY}^k) = \int_{\mathcal{X} \times \mathcal{Y}} d_{\mathcal{X}\mathcal{Y}}(x, y) \mathrm{d}\gamma^k(x, y). \tag{45}$$

Given $\nu_{XY} = \sum_{k=1}^{K} \lambda_k \nu_{XY}^k$, let $\gamma^* = \sum_{k=1}^{K} \lambda_k \gamma^k$. Since the first marginal of $\gamma^*$ is $\mu_{XY}$ and the second marginal of $\gamma^*$ is $\sum_{k=1}^{K} \lambda_k \nu_{XY}^k$, it follows that $\gamma^* \in \Gamma(\mu_{XY}, \nu_{XY})$. However, $\gamma^*$ is not necessarily optimal transport plan for $W(\mu_{XY}, \nu_{XY})$, we conclude that

$$W(\mu_{XY}, \nu_{XY}) = \inf_{\gamma \in \Gamma(\mu_{XY}, \nu_{XY})} \int_{\mathcal{X} \times \mathcal{Y}} d_{\mathcal{X}\mathcal{Y}}(x, y) \mathrm{d}\gamma(x, y) \leq \int_{\mathcal{X} \times \mathcal{Y}} d_{\mathcal{X}\mathcal{Y}}(x, y) \mathrm{d}\gamma^*(x, y)$$
$$= \sum_{k=1}^{K} \lambda_k \int_{\mathcal{X} \times \mathcal{Y}} d_{\mathcal{X}\mathcal{Y}}(x, y) \mathrm{d}\gamma^k(x, y) = \sum_{k=1}^{K} \lambda_k W(\mu_{XY}, \nu_{XY}^k). \tag{46}$$

$\square$

## D.5    PROOF OF THEOREM 5

*Proof.* Since the Wasserstein distance satisfies the triangle inequality, the distance $W(\mu_X, \nu_X)$ can be related to the empirical distributions $\widehat{\mu}_X$ and $\widehat{\nu}_X$ as follows:

$$W(\mu_X, \nu_X) \leq W(\widehat{\mu}_X, \mu_X) + W(\widehat{\mu}_X, \nu_X) \leq W(\widehat{\mu}_X, \mu_X) + W(\widehat{\mu}_X, \widehat{\nu}_X) + W(\widehat{\nu}_X, \nu_X). \tag{47}$$

Given $\mathbb{E}[W(\mu, \widehat{\mu}_X)] \leq \lambda_\mu n^{-1/\sigma_\mu}$ and $\mathbb{E}[W(\nu_X, \widehat{\nu}_X)] \leq \lambda_\nu m^{-1/\sigma_\nu}$ from Theorem 4, with probabilities at least $1 - e^{-2nt_\mu^2}$ and $1 - e^{-2mt_\nu^2}$, respectively, we have

$$W(\mu_X, \widehat{\mu}_X) \leq \lambda_\mu n^{-1/\sigma_\mu} + t_\mu;$$
$$W(\nu_X, \widehat{\nu}_X) \leq \lambda_\nu m^{-1/\sigma_\nu} + t_\nu. \tag{48}$$

It is reasonable to assume the two events in Eq. (48) are independent, so we can apply them to Eq. (47), and thus obtain

$$W(\mu_X, \nu_X) - W(\widehat{\mu}_X, \widehat{\nu}_X) \leq \lambda_\mu n^{-1/\sigma_\mu} + \lambda_\nu m^{-1/\sigma_\nu} + t_\mu + t_\nu \tag{49}$$

with probability at least $(1 - e^{-2nt_\mu^2})(1 - e^{-2mt_\nu^2})$.

Since $\mathbb{E}[W(\mu, \widehat{\mu}_X)]$ and $\mathbb{E}[W(\nu_X, \widehat{\nu}_X)]$ are non-negative, it follows that $\lambda_\mu, \lambda_\nu \geq 0$. Given that $t_\mu$ and $t_\nu$ are also positive, the right-hand side of Eq. (49) is non-negative. Therefore, we can take the absolute value on both sides of Eq. (49) without changing the direction of the inequality, leading to Eq. (30). $\square$

## E    DEMONSTRATION OF IMPLICIT DEPENDENCY

We demonstrate that $f_{\theta_Y}^{-1}(\overline{Y}_{n+1})$ implicitly depends on $X_{n+1}$ through the composition $\phi \circ f_{\theta_X}$, where $\phi : \mathcal{X} \to \mathcal{Y}$ is the ground truth mapping function under the calibration distribution $P_{XY}$. Consider a BNF $f_\theta$ is optimized by Wasserstein distance minimization in Eq. (16) such that $f_{\theta\#}Q_{XY} = P_{XY}$. Therefore, for a test sample $(X_{n+1}, Y_{n+1}) = (x, y) \sim Q_{XY}$, it holds that

$$(\overline{X}_{n+1}, \overline{Y}_{n+1}) = f_\theta(x, y) = (f_{\theta_X}(x), f_{\theta_Y}(y)) = (\overline{x}, \overline{y}) \sim P_{XY}.$$

As a result, $C_\mathrm{A}(\overline{X}_{n+1})$ satisfies the conditional coverage guarantee under $P_{XY}$. Moreover, since $\overline{y} = \phi(\overline{x})$, we obtain

$$f_{\theta_Y}^{-1}(\overline{y}) = f_{\theta_Y}^{-1}(\phi(\overline{x})) = f_{\theta_Y}^{-1}(\phi(f_{\theta_X}(x))),$$

which shows that the inverse transformation $f_{\theta_Y}^{-1}(\overline{y})$ used to construct $C_\mathrm{BNF}(X_{n+1})$ inherently captures the dependency on $X_{n+1} = x$.

We present an example to illustrate the dependency. Denote $\mathcal{U}$ and $\mathcal{N}$ uniform and Gaussian distributions, respectively. To introduce a distribution shift between $P_{XY}$ and $Q_{XY}$, let

$$P_X = \mathcal{U}(0,1), P_{Y|X} = \mathcal{N}(-0.5X, -0.3X^2 + 0.3X);$$
$$Q_X = \mathcal{U}(0,0.8), Q_{Y|X} = \mathcal{N}(0.25X, -0.24X^2 + 0.24X).$$

Figure 6 shows how the inverse transformation $f_{\theta_Y}^{-1}$ preserve the conditional guarantee from $C_A(\overline{X}_{n+1})$ to $C_{BNF}(X_{n+1})$ through the implicit dependency on $X_{n+1} = x$.

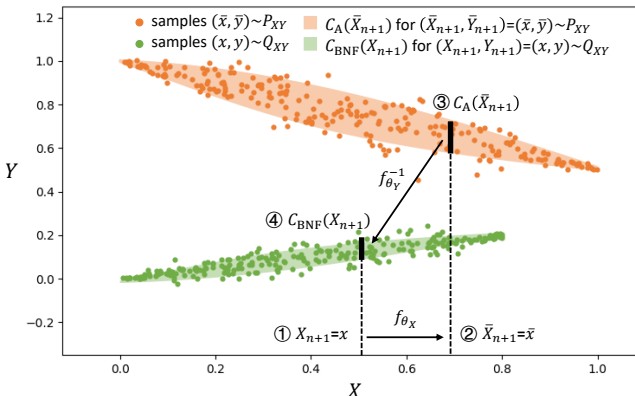

Figure 6: Preserving conditional coverage via implicit dependency on test input. The circled numbers indicate the sequential steps to obtain the corresponding values or prediction sets.

## F    COMPARISON BETWEEN NORMALIZING FLOW TECHNIQUES

The monotonicity of the univariate $f_{\theta_Y}$ allows us to take advantage of Proposition 6 to inversely transform $C_A(\overline{X}_{n+1})$ via Eq. (19). However, the monotonicity also limits the flexibility of $f_{\theta_Y}$, restricting the class of distributions it can model. Here, we briefly introduce several normalizing flow techniques designed for one-dimensional transformations that often struggle to map complex distributions effectively, thereby motivating the design of Augmented BNF in Section 4.2. For a more comprehensive overview of normalizing flows, we refer to the survey by Kobyzev et al. (2020).

We begin with planar flow, a fundamental transformation that expands or contracts the input space along specific directions (Rezende & Mohamed, 2016). A planar flow is achieved by applying a linear transformation followed by a nonlinear activation, which dictates how the data is warped. To enhance expressiveness, normalizing flows are typically constructed as compositions of multiple sub-flows. We implement a BNF where each branch applies a sequence of 16 planar flows. LeakyReLU is used as the nonlinear activation function to preserve invertibility throughout the transformation.

Residual flow is built using residual connections (He et al., 2015). The output of a residual connection is the sum of the original input and a transformation generated by a neural network. For these residual connections to be invertible, the transformation must have a Lipschitz constant less than 1, ensuring that the transformation does not distort the data too much. We also construct a BNF where each branch consists of 16 residual connections. The neural network within each residual connection has an architecture consisting of an input layer, two hidden layers with 128 units each, and an output layer matching the input dimension.

Both planar flow and residual flow are capable of transforming one-dimensional data. In addition, autoregressive flow (Kingma et al., 2016; Papamakarios et al., 2021) offers an alternative approach by modeling each transformation step as conditioned on the preceding ones—meaning the transformation of each sample value explicitly depends on the values that came before it. This sequential dependency enables more flexible and expressive density estimation, particularly in one-dimensional settings. However, because BNF requires deterministic transformations that are independent of input ordering, autoregressive flow is not suitable for our approach.

We illustrate the performance of BNFs constructed using planar flow and residual flow in Figure 7 and compare them against the Augmented BNF, which is implemented using a standard coupling normalizing flow, Real NVP (Dinh et al., 2016). Detailed specifications for the Augmented BNF are provided in Appendix H. The results show that BNFs using univariate $f_{\theta_Y}$ struggle to transform complex distributions effectively, resulting in higher ASCG compared to the Augmented BNF.

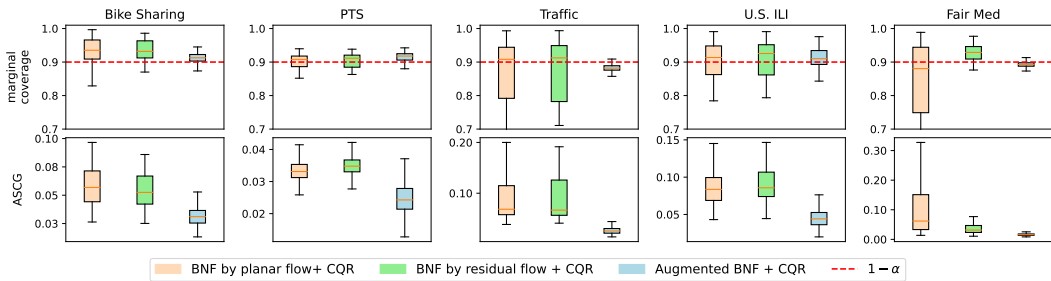

Figure 7: Marginal coverage and ASCG of BNFs constructed with planar and residual flows, compared with Augmented BNF at confidence level $1 - \alpha = 0.9$.

## G    EXACERBATED CURSE OF DIMENSIONALITY VIA FEATURE CONDITIONING

Feature-conditioned BNF, denoted by

$$f_\theta^{\text{fea}}(x, y) = \big(f_{\theta_X}(x), f_{\theta_Y}^{\text{fea}}(y; x)\big) = (\bar{x}, \bar{y}), \tag{50}$$

conditions the $Y$ transformations on input features, while keeping $f_{\theta_X}$ unchanged. This design theoretically improves the model's expressiveness and captures stronger dependencies between $X$ and $Y$. However, it increases the input dimension of $f_{\theta_Y}^{\text{fea}}$ to $d + 1$, making the total input dimension of Feature-conditioned BNF $2d + 1$. As a result, the curse of dimensionality is exacerbated with a small sample-to-dimension ratio $|\mathcal{S}_{D^k}|/(2d + 1)$, making true distributions harder to estimate. In contrast, Augmented BNF maintains a more favorable ratio of $|\mathcal{S}_{D^k}|/(d + 2)$, as the input to $f_{\theta_Y}^{\text{aug}}$ is only two-dimensional. Consequently, with limited data, Feature-conditioned BNF tends to yield higher ASCG due to poor approximation.

Table 1: Feature-conditioned BNF holds a **small** sample-to-dimension ratio $|\mathcal{S}_{D^k}|/(2d + 1)$.

| Dataset | Bike Rental | PTS | Traffic | U.S. ILI | Fair Med |
|---|---|---|---|---|---|
| $d$ | 4 | 9 | 3 | 2 | 2 |
| $|\mathcal{S}_{D^k}|$ | 2800 | 7500 | 2800 | 870 | 3000 |
| $|\mathcal{S}_{D^k}|/(d + 2)$ | 466.7 | 681.8 | 560.0 | 217.5 | 750.0 |
| $|\mathcal{S}_{D^k}|/(2d + 1)$ | **311.1** | **394.7** | **400.0** | **174.0** | **600.0** |

We report the small sample-to-dimension ratio of Feature-conditioned BNF in Table 1, and demonstrate its less robust conditional coverage in Figure 8, where its ASCG tend to be higher than that of Augmented BNF.

While projecting the original input $x$ of $f_{\theta_Y}^{\text{fea}}(y; x)$ in Eq. (50) to a one-dimensional representation $\tilde{x}$ (e.g., via PCA (Abdi & Williams, 2010), t-SNE (Van der Maaten & Hinton, 2008), or UMAP (McInnes et al., 2018)) can alleviate the curse of dimensionality, this dimensionality reduction inevitably discards information that may be crucial for accurately modeling the conditional distribution $P_{Y|x}$ of the calibration data. Consequently, $P_{Y|\tilde{x}}$ may fail to capture key dependencies in the true $P_{Y|x}$, limiting the effectiveness of conditioning the $Y$-branch on $\tilde{x}$.

In contrast, the augmented $Y$-branch $f_{\theta_Y}^{\text{aug}}(y; \varepsilon)$ in Augmented BNF can be viewed as conditioning on a simple one-dimensional Gaussian noise variable $\varepsilon$. Compared to $\varepsilon$, the projected feature $\tilde{x}$ lacks sufficient stochasticity or variability to provide the model with the expressive flexibility needed to capture a broad family of distributions. As a deterministic and compressed summary of $x$, $\tilde{x}$ is neither as informative as the original input nor as adaptable as a random noise input. As a result, conditioning

on $\tilde{x}$ is disadvantaged—it inherits neither the full structure of $x$ nor the modeling freedom enabled by stochastic conditioning on $\varepsilon$, as demonstrated in Figure 8.

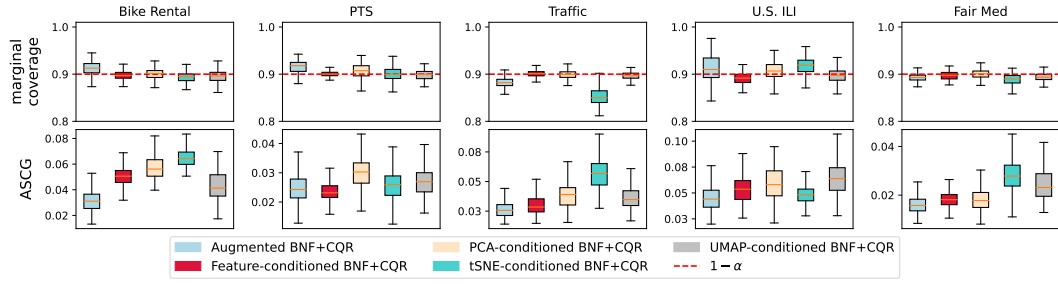

Figure 8: Performance comparison of conditioning the $Y$-branch on feature and its one-dimensional projection obtained via PCA, t-SNE, and UMAP.

## H STRUCTURE OF AUGMENTED BNF VIA COUPLING FLOWS

Both branches of Augmented BNF operate on multi-dimensional data, enabling the use of coupling flows—a technique for modeling complex high-dimensional distributions. A coupling flow usually consists of multiple coupling layers. In a coupling layer, the input is partitioned into two parts. One part remains unchanged during the transformation, while the other is modified using a neural network $c$, whose parameter $\Theta$ depends on the unchanged part. This setup ensures invertibility and allows for flexible, learnable transformations. Afterward, a permutation step is applied for higher expressiveness. In our implementation, each branch of Augmented BNF consists of a sequence of 48 coupling layers based on Real NVP (Dinh et al., 2016), allowing the entire input to be progressively transformed. The neural network $c$ within each coupling layer follows a symmetric architecture with hidden layers of sizes 64, 128, 256, 128, and 64, mapping from the input dimension to the output dimension. Figure 9 illustrates the structure of a coupling layer, using a random variable $Z \in \mathbb{R}^d$ with a realization $z$, and shows how both branches are constructed by stacking multiple coupling layers. The normalized Gaussian noise $\bar{\varepsilon}$ is discarded after the transformation.

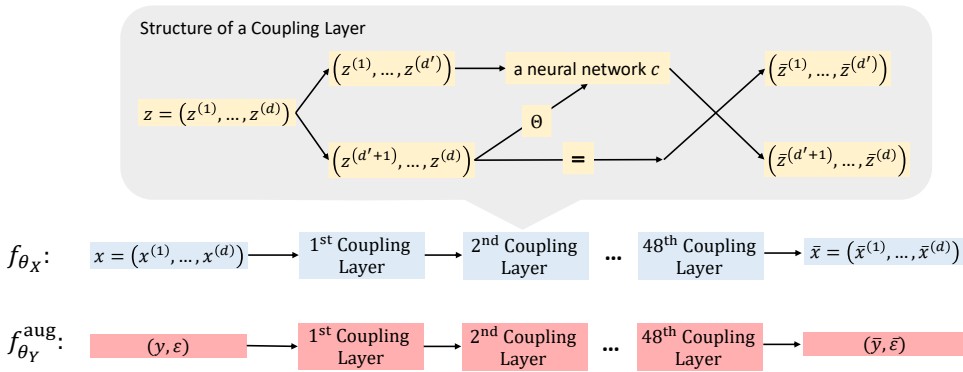

Figure 9: Illustration of the coupling layer structure and the overall composition of Augmented BNF.

## I CONFORMALIZED QUANTILE REGRESSION

Conformalized quantile regression (CQR) (Romano et al., 2019) first trains two regression models with pinball loss at levels $1 - \alpha/2$ and $\alpha/2$, respectively, then calibrates the resulting intervals using residuals on a separate calibration set. Under the assumption that test and calibration samples are i.i.d., the calibrated intervals ensure conditional coverage with finite samples.

For clarity, we introduce CQR in the context of sample normalization and multi-source domain generalization. For a regression model $h$, the pinball loss (Steinwart & Christmann, 2011) at quantile

level $\alpha$ for sample $(x, y)$ is defined as

$$l_\alpha(h(x), y) = \begin{cases} \alpha\,(y - h(x)) & \text{if } y - h(x) > 0, \\ (1 - \alpha)\,(h(x) - y) & \text{otherwise.} \end{cases} \tag{51}$$

We denote $\mathcal{S}_D = \bigcup_{k=1}^K \mathcal{S}_{D^k}$ the union of all training sets. The models $h_{\text{hi}}$ and $h_{\text{lo}}$ are obtained by optimizing the pinball loss in Eq (51) at quantile levels $1 - \alpha/2$ and $\alpha/2$ on $\mathcal{S}_D$, respectively. For calibration instances $\{(X_i, Y_i)\}_{i=1}^n$ drawn from $P_{XY}$, conformal scores are defined as

$$V_i = \max\{h_{\text{lo}}(X_i) - Y_i, Y_i - h_{\text{hi}}(X_i)\}, \text{ for } i = 1, ..., n. \tag{52}$$

Let $\tau$ be the $\lceil (1-\alpha)(n+1) \rceil / n$ quantile of $\{V_i\}_{i=1}^n$. If a test sample $(X_{n+1}, Y_{n+1}) \sim Q_{XY}$ is normalized to $(\overline{X}_{n+1}, \overline{Y}_{n+1}) \sim P_{XY}$, we construct an adaptive prediction set

$$C_{\text{CQR}}(\overline{X}_{n+1}) = \left[h_{\text{lo}}(\overline{X}_{n+1}) - \tau, h_{\text{hi}}(\overline{X}_{n+1}) + \tau\right]. \tag{53}$$

Here, $h_{\text{lo}}$ and $h_{\text{hi}}$ predict the likely lower and upper ends, while $\tau$ adjusts the set based on how well the predictions fit the calibration data. As proved in (Romano et al., 2019), $C_{\text{CQR}}$ can empirically approximate the conditional coverage guarantee described in Eq. (17). Extensions of CQR are explored in (Kivaranovic et al., 2020; Sesia & Candès, 2020), which modified the score function in Eq. (52) for higher adaptiveness.

## J  A BRIEF REVIEW OF THE SINKHORN ALGORITHM

As we introduced in Definition 3, the Wasserstein distance between two empirical distributions $\widehat{\mu}_X$ and $\widehat{\nu}_X$ with $p = 1$ is given by

$$W(\widehat{\mu}_X, \widehat{\nu}_X) = \min_{\gamma \in \Gamma(\widehat{\mu}_X, \widehat{\nu}_X)} \sum_{i=1}^n \sum_{j=1}^m \gamma_{ij}\, C_{ij}.$$

where $C \in \mathbb{R}^{n \times m}$ is the cost matrix with entries $C_{ij} = d_{\mathcal{X}}(x_i, x_j')$, and $\Gamma(\widehat{\mu}_X, \widehat{\nu}_X)$ is the set of joint distributions $\gamma \in \mathbb{R}_+^{n \times m}$ with marginals $\widehat{\mu}_X$ and $\widehat{\nu}_X$.

To make this optimization problem more tractable, the Sinkhorn algorithm (Cuturi, 2013) introduces an entropic regularization term:

$$W^\beta(\widehat{\mu}_X, \widehat{\nu}_X) = \min_{\gamma \in \Gamma(\widehat{\mu}_X, \widehat{\nu}_X)} \sum_{i=1}^n \sum_{j=1}^m \gamma_{ij}\, C_{ij} + \beta \sum_{i=1}^n \sum_{j=1}^m \gamma_{ij} \log \gamma_{ij},$$

where $\beta > 0$ controls the strength of the regularization.

This regularized objective is strictly convex and can be efficiently minimized via iterative matrix scaling. Let $K = \exp(-C/\beta)$ be the Gibbs kernel. The scaling vectors $u \in \mathbb{R}^n$ and $v \in \mathbb{R}^m$ are initialized to all ones and updated via

$$u \leftarrow \frac{1/n}{Kv}, \quad v \leftarrow \frac{1/m}{K^\top u},$$

where divisions are element-wise. Once converged with small changes in $u$ and $v$, the optimal transport plan takes the form

$$\gamma^* = \text{diag}(u)\, K\, \text{diag}(v).$$

This approach yields a differentiable approximation to the true Wasserstein distance, enabling its integration into gradient-based optimization pipelines. We refer to (Cuturi, 2013; Knight, 2008; Feydy, 2020) for more detailed studies about the Sinkhorn algorithm.

## K  INTRODUCTION TO BASELINES

Figure 10 highlights the distinctions between the baseline methods and the proposed approach. SCP constructs prediction sets of fixed size and ensures only marginal coverage under i.i.d. assumptions, rendering it ineffective under joint distribution shifts. IW-CP addresses only covariate shift and

causes its prediction intervals to contract in the example, because test features are distributed in regions where calibration data is concentrated. WC-CP accounts for worst-case distribution shifts, expanding prediction sets until $1 - \alpha$ marginal coverage is achieved on the test data, which can be inefficient. WR-CP improves upon this by regularizing the base predictive model through minimizing the Wasserstein distance between calibration and test conformal scores, producing more compact prediction sets while maintaining robust marginal coverage. All of these methods, however, focus exclusively on marginal coverage. CQR, a representative conditional conformal prediction method, fails to handle distributional shifts. In contrast, the proposed Augmented BNF transformation model learns an invertible mapping between calibration and test data, enabling robust conditional coverage even under non-i.i.d. conditions.

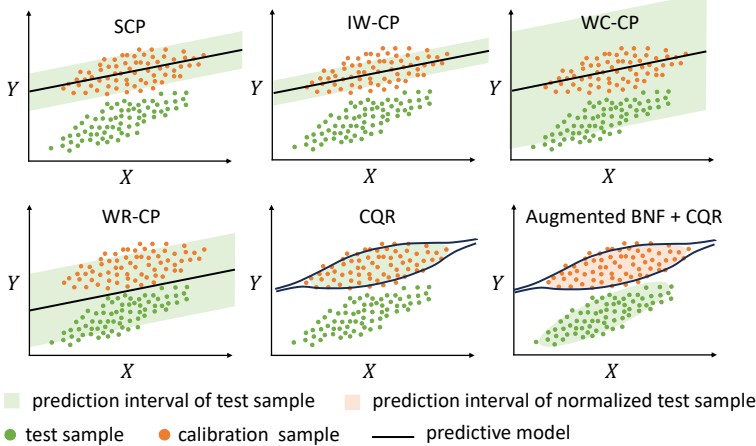

Figure 10: Comparison between baselines and the proposed method via a toy example. Augmented BNF effectively transforms the prediction intervals from the calibration distribution to the test distribution.

## L  INTRODUCTION TO DATASETS

### L.1  DATA PREPARATION

We introduce the data preparation procedure shared across all datasets. We set $K = 3$, partitioning each dataset into three subsets, each exhibiting a distinct distribution shift. For each dataset, we conduct 10 independent sampling trials. In each trial, we first sample $\mathcal{S}_{D^k}$ from subset $k$ without replacement. Since calibration and training data typically share the same distribution in conformal prediction, $\mathcal{S}_P$ is then sampled from the union of all $K$ subsets, also without replacement. Finally, 100 different $\mathcal{S}_Q$ sets are sampled as random mixtures from the remaining data. This procedure ensures that $\mathcal{S}_{D^k}$ for $k = 1, \ldots, K$, $\mathcal{S}_P$, and $\mathcal{S}_Q$ are mutually disjoint. Since the Sinkhorn algorithm is more numerically stable when comparing empirical distributions with matching sample sizes, we set the calibration set and each training set to have equal sizes, i.e., $|\mathcal{S}_P| = |\mathcal{S}_{D^k}|$ for all $k = 1, \ldots, K$. Experimental results are aggregated over the 10 trials for each dataset.

We also leverage a toy example from (Xu et al., 2025) to demonstrate joint distribution shift under multi-source domain generalization in Figure 11.

### L.2  SYNTHETIC DISTRIBUTION SHIFTS

The Physicochemical Properties of Protein Tertiary Structure (PTS) dataset (Rana, 2013) contains 45,730 instances, with the target variable being the protein decoy size. It includes nine features: surface area, non-polar exposed area, fractional area of exposed non-polar residue, fractional area of exposed non-polar part, molecular mass weighted exposed area, average deviation, Euclidean distance, secondary structure penalty, and spatial distribution constraints. Raw data is split into three subsets based on the distribution of the secondary structure penalty, thereby introducing distribution shifts among the subsets. We also use the PTS dataset to perform ablation studies on the approximation

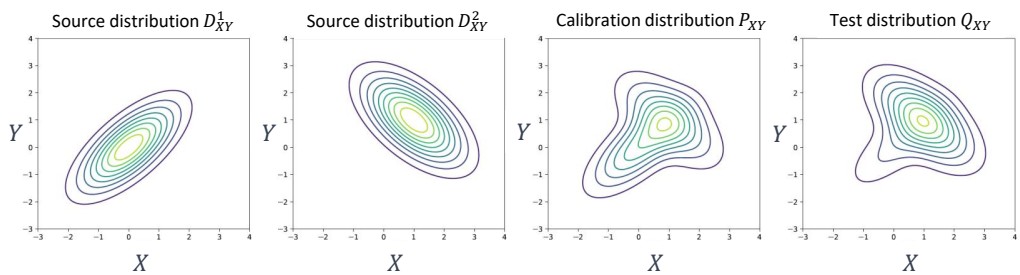

Figure 11: **Joint distribution shift under multi-source domain generalization.** The test distribution $Q_{XY}$ is a random mixture of source distributions, while the calibration distribution $P_{XY}$ is a fixed known mixture. As a result, joint distribution shift occurs since $P_{XY} \neq Q_{XY}$.

ability (with varying sample sizes) and generalization performance (with different numbers of source domains) of Augmented BNF.

### L.3 NATURAL DISTRIBUTION SHIFTS REFLECTING REAL-WORLD CHALLENGES

**Generalized sales prediction over time-series** is crucial for risk-averse business decision-making (Jin et al., 2022). Moreover, sales data typically exhibit strong periodic patterns, such as seasonal or weekly fluctuations. Thus, effectively utilizing data from each sub-period to model a robust and generalized sales pattern is critical for achieving reliable forecasts. This requires models not only to capture short-term variations but also to generalize across different temporal domains, where distribution shifts may occur naturally due to changes in consumer behavior, external events, or market conditions. We consider the Bike Rental dataset (Fanaee-T, 2013) to reflect this challenge. The dataset records hourly and daily rental counts from the Capital Bikeshare system during 2011 and 2012, along with associated weather and seasonal information. We partition the data based on rental hours into three time intervals: [0,8] (midnight), [9,16] (daytime), and [17,23] (evening). For prediction, we select continuous features including temperature, feeling temperature, humidity, and wind speed. The target variable is the count of rental bikes.

**Traffic speed prediction with mismatched data** focuses on transferring models trained on source distributions (e.g., traffic patterns on regular days and at major intersections) to test distributions exhibiting different characteristics (e.g., traffic patterns on special days and at minor intersections). For example, recent work has proposed traffic-law-informed models based on reaction-diffusion equations to provide generalized speed predictions (Sun et al., 2023). Nevertheless, enhancing the reliability of uncertainty quantification under such distribution shifts remains a significant challenge. The Seattle-Loop dataset contains traffic volume and speed data collected in Seattle throughout 2015, recorded by sensors at 5-minute intervals (Cui et al., 2019). PEMSD4 includes traffic data from 29 roads in San Francisco collected between January and February 2018, while PEMSD8 covers 8 roads in San Bernardino from July to August 2016 (Bai et al., 2020). The task is to predict traffic speed at the next time step based on current speed and volume measurements. With $K = 3$, we select one representative intersection from each dataset. Due to varying local traffic patterns, natural distribution shifts arise among the three locations. Our goal is to achieve strong generalization across these locations, ensuring robust predictions on any test sites where traffic patterns resemble a random mixture of the three selected intersections.

**Fair medical decision-making for patients from different hospitals** is essential for ensuring equitable healthcare outcomes. Variations in patient demographics, medical imaging scanners, laboratory equipment, and clinical practices across hospitals can lead to distribution shifts in the data. This phenomenon is commonly referred to as the multi-center issue (Das, 2022). Addressing this challenge is essential for building predictive models that remain accurate and fair across diverse healthcare institutions (Olsson et al., 2022). To validate the effectiveness of the proposed method in this task, we collect patient data from a collaborating hospital. Additionally, we use the MIMIC-IV (Johnson et al., 2023) and eICU (Pollard et al., 2018) datasets to simulate data from two other hospitals. The goal is to fairly predict patients' ICU stay times based on their Apache scores and blood urea nitrogen (BUN) levels, ensuring reliable performance regardless of which center a patient

originates from. We consider fair medical prediction to be achieved across the three data sources if the model exhibits comparable performance on random mixtures of the sources.

**Robust epidemic modeling across pandemic phases** can facilitate timely public health responses and resource planning. The U.S. Centers for Disease Control and Prevention (CDC) categorizes an epidemic period into three main phases: initiation, acceleration, and deceleration (CDC). Each of these phases exhibits distinct epidemiological characteristics, which lead to natural distribution shifts. Traditional forecasting methods typically rely on Susceptible-Infectious-Recovered (SIR) models (Harko et al., 2014; Kabir et al., 2019; Turkyilmazoglu, 2022) to predict the number of recently infected patients, aiming for robustness across the different pandemic phases. We demonstrate the application of the proposed method using the U.S. Influenza-like Illness (ILI) dataset (Deng et al., 2020), which contains weekly reports from the CDC on the number of ILI patients. The objective is to predict new infections for the upcoming week using both the weekly increase of infected patients and the cumulative infections for the year. The raw data is divided into three subsets based on the corresponding pandemic phases. We consider a forecasting model to be robust if its predictions remain reliable on random mixtures of data from the three phases.

We further apply t-SNE (Van der Maaten & Hinton, 2008) to map the samples from each source into two dimensions, as shown in Figure 12. The visualization reveals clear distributional shifts between most sources. However, for some cases, such as the second and third sources in the Bike Rental and Fair Med setups, the distributions appear more similar. This slight overlap is not the result of manually creating similar data but arises naturally from the datasets themselves, which are collected from real-world scenarios.

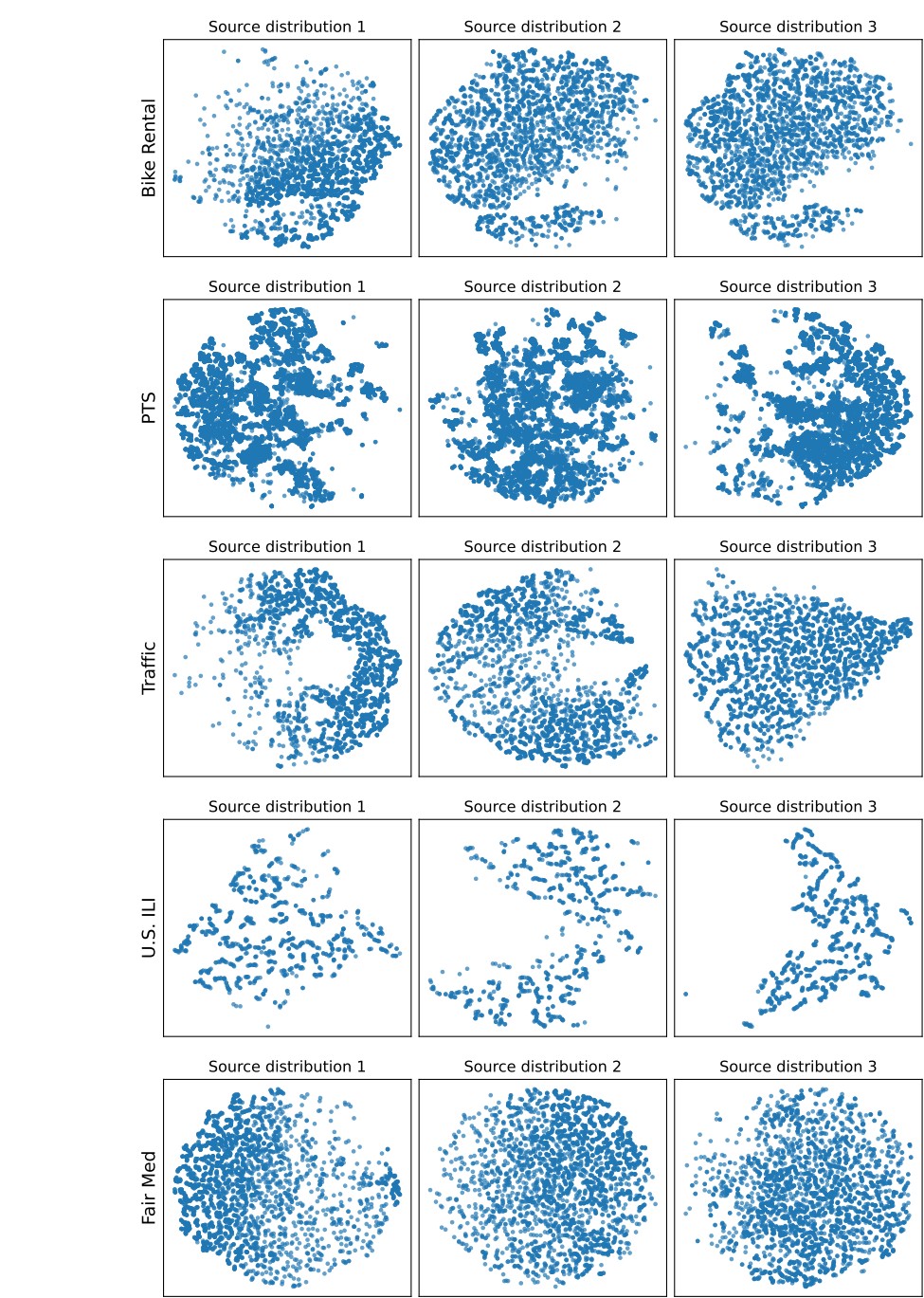

Figure 12: Empirical data distributions of source domains after applying t-SNE.

# M CONDITIONAL COVERAGE EVALUATION METRIC

## M.1 WORST-SLICE COVERAGE (WSC)

Worst-slice coverage (WSC) (Cauchois et al., 2021) quantifies the minimum empirical coverage over any slab $\mathcal{S} \subseteq \mathcal{X}$ that contains at least 10% of the test samples in $\mathcal{S}_Q$. Specifically, for any CP methods that produce a prediction set $C(x)$ given an input $x$, WSC is defined by

$$\text{WSC} = \inf_{\mathcal{S} \subseteq \mathcal{X}} \Pr(y \in C(x) | x \in \mathcal{S}), \text{s.t.} \Pr(x \in \mathcal{S} | (x, y) \in \mathcal{S}_Q) \geq 0.1. \tag{54}$$

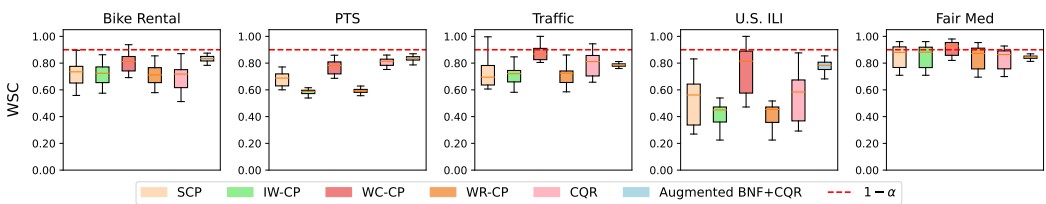

Figure 13: WSC of Augmented BNF+CQR and five baselines with $1 - \alpha = 0.9$.

Figure 13 reports the WSC performance of our method and baselines following the implementation of (Romano et al., 2020). Our approach consistently achieves WSC values close to the desired confidence level $1 - \alpha = 0.9$. Worst-Case CP (WC-CP) shows similarly high WSC, but a closer inspection reveals that this is largely driven by its conservative behavior, which produces substantial over-coverage, as illustrated in Figure 10.

This discrepancy arises because WSC, defined in Eq. (54), only evaluates the infimum slice coverage and therefore fails to penalize over-coverage. As also noted in (Romano et al., 2020), ensuring a high worst-case slice does not guarantee good conditional coverage across $\mathcal{X}$, particularly when different regions exhibit excessive coverage.

These limitations motivate our introduction of ASCG, which evaluates a richer family of subsets (Eq. (27)) and penalizes both under- and over-coverage from the target level $1 - \alpha$. As a result, ASCG provides a more comprehensive and balanced assessment of conditional coverage robustness.

### M.2 WORST-SLICE COVERAGE GAP (WSCG)

We propose a variant of WSC called worst-slice coverage gap (WSCG), quantified by

$$\text{WSCG} = \sup_{\mathcal{S} \subseteq \mathcal{X}} |\Pr(y \in C(x) | x \in \mathcal{S}) - (1 - \alpha)|, \text{s.t.} \Pr(x \in \mathcal{S} | (x, y) \in \mathcal{S}_Q) \geq 0.1. \quad (55)$$

By taking the maximum absolute difference from the desired confidence level $1 - \alpha$, WSCG captures both under-coverage and over-coverage in any sufficiently large slice of the input space. This makes it a more stringent and informative metric than WSC.

As illustrated in Figure 14, when $1 - \alpha = 0.9$, our method consistently achieves low WSCG values across datasets, demonstrating its robustness in maintaining conditional coverage even in the most challenging regions. Interestingly, we observe that WC-CP consistently attains a WSCG of 0.1 on the Traffic setup. This occurs because its conservative nature makes the coverage on individual slabs ranges between 0.8 and 1.0 on the Traffic setup, preventing the WSCG from exceeding 0.1. The conservativeness becomes even more evident when we reduce the target coverage to $1 - \alpha = 0.8$ in Figure 15. In that case, WC-CP has additional slack, and its WSCG correspondingly increases to 0.2.

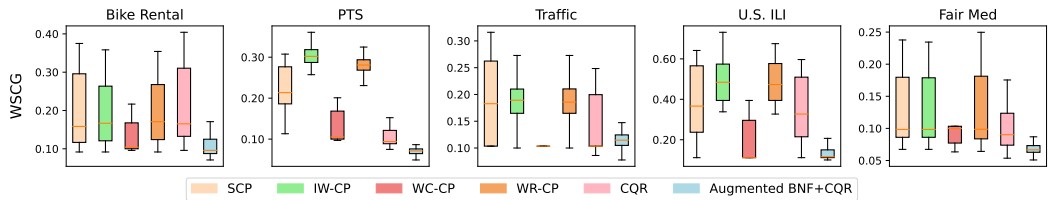

Figure 14: WSCG of Augmented BNF+CQR and five baselines with $1 - \alpha = 0.9$.

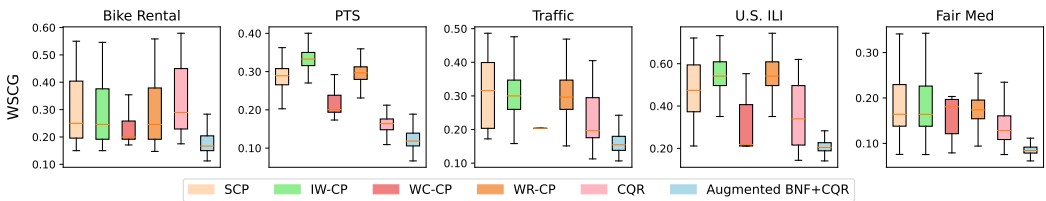

Figure 15: WSCG of Augmented BNF+CQR and five baselines with $1 - \alpha = 0.8$.

### M.3 MULTIVARIATE AVERAGE-SLICE COVERAGE GAP

While ASCG introduced in Eq. (28) computes coverage gaps only along one-dimensional slices, it does not fully capture potential failures in multivariate regions of the input space. Inspired by WSCG, we introduce a multivariate version of ASCG defined as

$$
\text{Multivariate ASCG} = \frac{1}{N_{\mathcal{S}}} \sum_{\mathcal{S} \subseteq \mathcal{X}} \left| \Pr(y \in C(x) \mid x \in \mathcal{S}) - (1 - \alpha) \right|,
$$

$$
\text{s.t. } \Pr(x \in \mathcal{S} \mid (x, y) \in \mathcal{S}_Q) \geq 0.1,
$$

where $N_{\mathcal{S}}$ is the total number of multivariate slabs considered. We follow (Romano et al., 2020) by randomly generating 1000 slabs, and therefore set $N_{\mathcal{S}} = 1000$. This multivariate ASCG generalizes the original ASCG metric by evaluating and aggregating coverage gaps across subsets formed in multivariate partitions of the feature space. As shown in Figure 16, our method maintains low multivariate ASCG values, demonstrating reliable conditional coverage even on complex multivariate partitions of the test space.

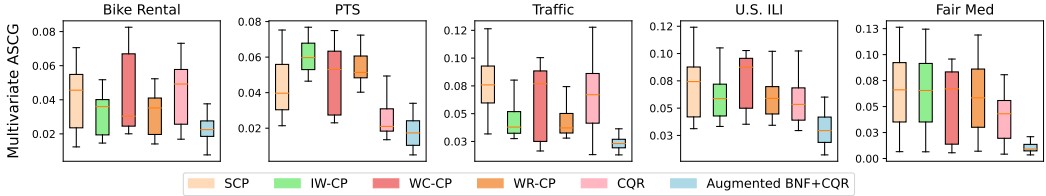

Figure 16: Multivariate ASCG of Augmented BNF+CQR and five baselines with $1 - \alpha = 0.9$.

## N GENERALIZATION PERFORMANCE OF AUGMENTED BNF

### N.1 VARIOUS $K$ VALUES

To explore the generalization ability of Augmented BNF under varying numbers of source domains, we modified the sampling procedure in Appendix L by changing $K \in \{2, 3, 4, 8, 12\}$. For each value of $K$, we generated 10 independent trials using the PTS dataset to account for sampling variability. Augmented BNF combined with CQR was applied to each trial across confidence levels $1 - \alpha \in [0.1, 0.9]$, enabling a comprehensive evaluation. Figure 17 shows that increasing the number of source domains does not significantly degrade conditional coverage robustness, suggesting that Augmented BNF generalizes well even in the presence of greater domain heterogeneity.

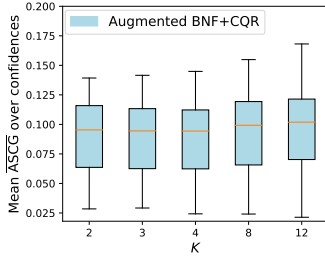

Figure 17: Generalization performance of Augmented BNF+CQR with different numbers of source domains.

### N.2 DIFFERENT SAMPLE SIZES

Generative models may struggle to approximate underlying distributions when data are limited, especially in high dimensions (Kong & Chaudhuri, 2020; Poggio et al., 2017). To assess how sample

size affects the performance of Augmented BNF, we vary the number of training samples and perform 10 trials for each setting on the PTS dataset. For each trial, we apply Augmented BNF+CQR across $1-\alpha$ from 0.1 to 0.9 and compute the mean $\overline{\text{ASCG}}$ over all confidence levels. As shown in Figure 18, the results reveal a clear trend: conditional coverage becomes more robust as the size of each training set increases.

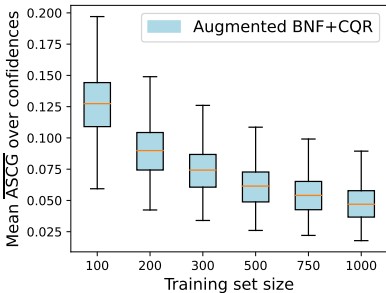

Figure 18: Impact of data availability on Augmented BNF approximation ability.

## O  COVERAGE LOWER BOUNDS UNDER IMPERFECT TRANSFORMATION

### O.1  MARGINAL COVERAGE LOWER BOUND

We establish a marginal coverage lower bound by quantifying the alignment between the calibration conformal scores and those obtained from the transformed test distribution.

Marginal coverage gap can be defined as the discrepancy between the CDFs of $P_V$ and $Q_V$ at the calibration quantile $\tau$ (Xu et al., 2025). After applying the transformation $f_\theta^{\text{aug}}$, the test distribution $Q_{XY}$ is mapped to $f_{\theta}^{\text{aug}}{}_{\#}Q_{XY}$, yielding the conformal score distribution $s_{\#}(f_{\theta}^{\text{aug}}{}_{\#}Q_{XY})$, where $s$ denotes the score function. The residual marginal coverage gap after transformation is therefore

$$|F_{P_V}(\tau) - F_{s_{\#}(f_{\theta}^{\text{aug}}{}_{\#}Q_{XY})}(\tau)|, \tag{57}$$

with $F$ denoting the CDF of the distribution indicated in the subscript.

This leads to the following lower bound on the marginal coverage of prediction sets produced by the Augmented BNF transformation model:

$$\Pr\big(Y_{n+1} \in C_{\text{BNF}}^{\text{aug}}(X_{n+1})\big) \geq 1 - \alpha - |F_{P_V}(\tau) - F_{s_{\#}(f_{\theta}^{\text{aug}}{}_{\#}Q_{XY})}(\tau)|. \tag{58}$$

Within the multi-source domain generalization (MSDG) framework, the test distribution $Q_{XY}$ is assumed to be a random mixture of source distributions $D_{XY}^{k}{}_{k=1}^{K}$. In this setting, we can bound the coverage gap as

$$|F_{P_V}(\tau) - F_{s_{\#}(f_{\theta}^{\text{aug}}{}_{\#}Q_{XY})}(\tau)| \leq \sup_{k\in\{1,...,K\}} |F_{P_V}(\tau) - F_{s_{\#}(f_{\theta}^{\text{aug}}{}_{\#}D_{XY}^{k})}(\tau)|. \tag{59}$$

Consequently, we obtain the final marginal coverage lower bound under MSDG as

$$\Pr\big(Y_{n+1} \in C_{\text{BNF}}^{\text{aug}}(X_{n+1})\big) \geq 1 - \alpha - \sup_{k\in\{1,...,K\}} |F_{P_V}(\tau) - F_{s_{\#}(f_{\theta}^{\text{aug}}{}_{\#}D_{XY}^{k})}(\tau)|. \tag{60}$$

For validation, we compare the theoretical bound with the empirical marginal coverage observed across randomly sampled test distributions. The results, presented in Figure 19, show that the empirical coverage for most test distributions exceeds the proposed lower bound, thereby confirming the validity of Eq. (60). The closeness between $1-\alpha$ and the bound further indicates that our method effectively aligns the calibration and test distributions.

### O.2  CONDITIONAL COVERAGE LOWER BOUND

Next, we establish a conditional coverage lower bound that accounts for the imperfect alignment between calibration and test data induced by Augmented BNF.

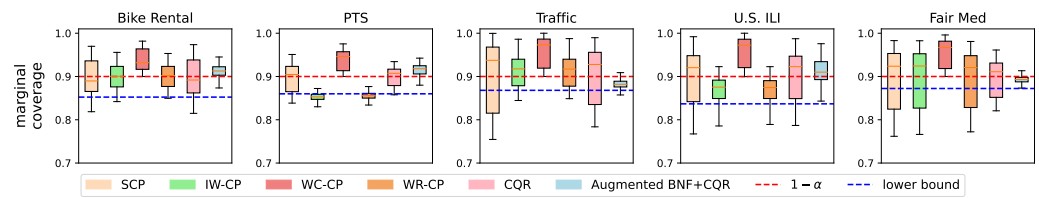

Figure 19: Empirical marginal coverage after transformation compared with the proposed lower bound.

First, even in the i.i.d. setting, exact conditional coverage is unattainable with finite samples (Vovk, 2012; Lei & Wasserman, 2014; Foygel Barber et al., 2021), as discussed in Section 2.1. For instance, Romano et al. (2019) explicitly note that CQR achieves conditional coverage only on the training data, not on unseen test samples. Likewise, the performance of LCP (Guan, 2023) is highly sensitive to the choice of kernel bandwidth, preventing finite-sample conditional coverage guarantees. Consequently, even under i.i.d. assumptions, the application of CQR can only guarantee that

$$\Pr(Y_{n+1} \in C_{\text{CQR}}(X_{n+1})|X_{n+1} = x) \geq 1 - \alpha - \alpha_{\text{i.i.d.}}, \tag{61}$$

where $\alpha_{\text{i.i.d.}}$ reflects the approximation error introduced by CQR. This gap is an intrinsic limitation of existing conditional CP approaches.

Secondly, under distribution shift, for a test sample $(X_{n+1}, Y_{n+1}) \sim Q_{XY}$, we have

$$F_{P_{V|x}} \left( \tau(x, V_i : X_i = x_{i=1}^n) \right) \geq 1 - \alpha, \tag{62}$$

$$F_{Q_{V|x}} \left( \tau(x, V_i : X_i = x_{i=1}^n) = \Pr\left(Y_{n+1} \in C_A(X_{n+1})|X_{n+1} = x\right). \tag{63}$$

As a result, the Conditional Coverage Gap (CCG) defined in Eq. (4) leads to

$$\Pr\left(Y_{n+1} \in C_A(X_{n+1})|X_{n+1} = x\right) \geq 1 - \alpha - \text{CCG}(P, Q, x) \tag{64}$$

Accounting for the approximation error $\alpha_{\text{i.i.d.}}$ from CQR, we can derive

$$\Pr\left(Y_{n+1} \in C_{\text{CQR}}(X_{n+1})|X_{n+1} = x\right) \geq 1 - \alpha - \alpha_{\text{i.i.d.}} - \text{CCG}(P, Q, x) \tag{65}$$

To evaluate the expected conditional coverage across the test distribution, we take the expectation over $x \sim Q_X$ and obtain

$$\mathbb{E}_{x \sim Q_X}[\Pr(Y_{n+1} \in C_{\text{CQR}}(X_{n+1})|X_{n+1} = x)] \geq 1 - \alpha - \alpha_{\text{i.i.d.}} - \text{ICG}(P, Q), \tag{66}$$

where the ICG$(P, Q)$ is defined in Eq. (5) as the expectation of CCG$(P, Q, x)$ over $Q_X$.

Finally, using our bound on ICG$(P, Q)$ in terms of the Wasserstein distance $W(P_{XY}, Q_{XY})$ in Eq. (13), we obtain a bound on the expected conditional coverage under distribution shift:

$$\mathbb{E}_{x \sim Q_X}[\Pr(Y_{n+1} \in C_{\text{CQR}}(X_{n+1})|X_{n+1} = x)] \geq 1 - \alpha - \alpha_{\text{i.i.d.}}$$
$$- \sqrt{2\kappa L} \left(\eta \cdot W(P_{XY}, Q_{XY}) + 1/4\right). \tag{67}$$

Finally, the transformation by Augmented BNF lead to a more robust prediction set $C_{\text{BNF}}^{\text{aug}}(X_{n+1})$. We clarify the role of the remaining Wasserstein distance $W(P_{XY}, f_{\theta\ \#}^{\text{aug}} Q_{XY})$ by

$$\mathbb{E}_{x \sim Q_X}[\Pr\left(Y_{n+1} \in C_{\text{BNF}}^{\text{aug}}(X_{n+1})|X_{n+1} = x\right)] \geq 1 - \alpha - \alpha_{\text{i.i.d.}}$$
$$- \sqrt{2\kappa L} \left(\eta \cdot W(P_{XY}, f_{\theta\ \#}^{\text{aug}} Q_{XY}) + 1/4\right),$$

where the term $\eta$ can be obtained by substituting $Q_{XY}$ and $Q_{Y|x}$ with $f_{\theta\ \#}^{\text{aug}} Q_{XY}$ and $f_{\theta_Y\ \#}^{\text{aug}} Q_{Y|x}$, respectively, in Eq. (13).

We denote $\alpha_{\text{trans}} = \sqrt{2\kappa L}(\eta \cdot W(P_{XY}, f_{\theta\ \#}^{\text{aug}} Q_{XY}) + 1/4)$ to quantify the remaining deviation induced by imperfect alignment between calibration and test distributions.

To evaluate the magnitude of $\alpha_{\text{trans}}$, we compare the Average Slice Coverage Gap (ASCG) across a range of confidence levels $1 - \alpha \in [10\%, 90\%]$, under the following three settings: (1) CQR under the distribution shift, (2) Augmented BNF+CQR under distribution shift, and (3) CQR under i.i.d. condition. Figure 20 shows that the proposed transformation model effectively approximates the CQR under the i.i.d. condition. This suggests that $\alpha_{\text{trans}}$ is significantly smaller than $\alpha_{\text{i.i.d.}}$ with the remaining coverage gap primarily attributable to the approximation error of CQR itself.

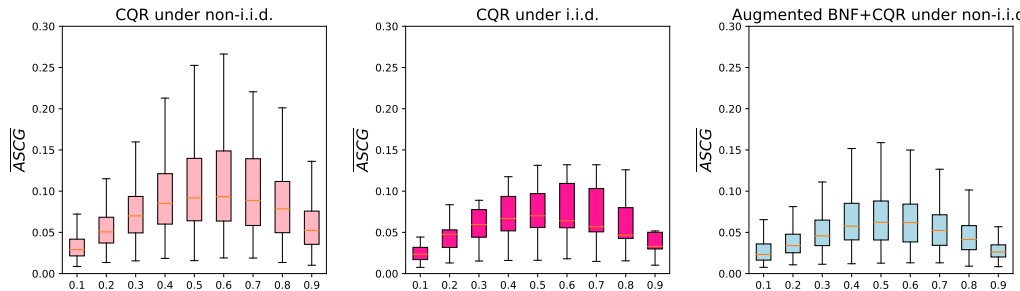

Figure 20: The transformation by Augmented BNF can effectively simulate the i.i.d. condition.

# P    PREDICTION EFFICIENCY UNDER SOURCE CONDITIONING

## P.1    PREDICTION INEFFICIENCY BY AUGMENTED BNF

Augmented BNF uses Eq. (22) to obtain prediction sets on the test distribution $Q_{XY}$. During training, this augmented component $\varepsilon$ of the $Y$ branch $f_{\theta_Y}^{\text{aug}}$ in Eq. (21) is sampled from a single Gaussian distribution $\mathcal{N}(0,1)$, making it independent of the training sample sources. As a result, the model learns a shared transformation for all training distributions $D_{XY}^k$ for $k = 1, ..., K$ to align with the calibration distribution $P_{XY}$.

At test time, this design leads to a key limitation: since $\varepsilon_{n+1}$ is source-agnostic, the $Y$ branch $f_{\theta_Y}^{\text{aug}}$ cannot infer which source distribution a new test sample originates from. As a result, the prediction set $C_{\text{BNF}}^{\text{aug}}(X_{n+1})$ must widen to account for all sources to ensure valid coverage. This behavior corresponds to a conditional worst-case strategy and inherently results in larger prediction sets.

The prediction inefficiency is reflected in Figure 21. The Augmented BNF produces noticeably larger prediction sets on Traffic, U.S. ILI, and Fair Med. This is due to the substantial variation in the conditional label distributions $D_{Y|x}^k$ within the three settings, such as significantly different supports, which leads to enlarged prediction sets.

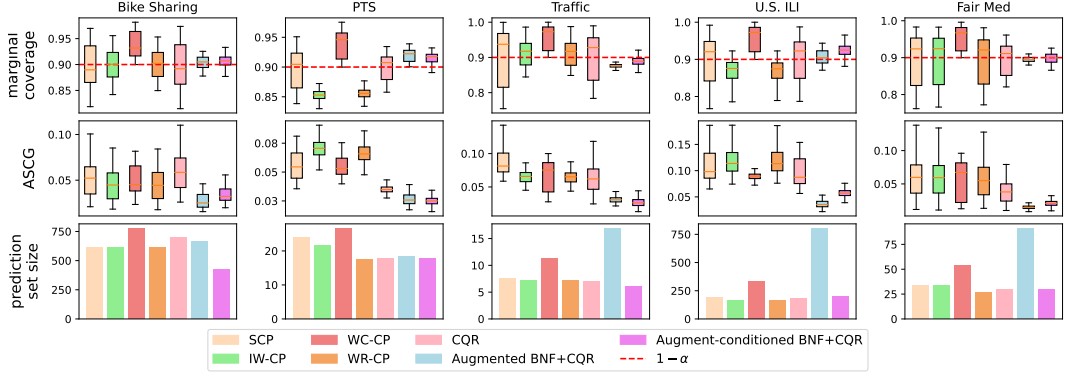

Figure 21: Prediction set size comparison. The standard Augmented BNF can produce large prediction sets, whereas the Augment-Conditioned variant significantly reduces set size while preserving coverage performance.

## P.2    EFFICIENT PREDICTION BY AUGMENT-CONDITIONED BNF

To achieve smaller prediction sets, we propose **Augment-conditioned BNF**, denoted as $f_{\theta}^{\text{aug-cond}}$. In this design, the augmented component $\varepsilon$ is sampled from a distinct Gaussian distribution $\mathcal{N}^k$ if an instance $(x, y)$ is from $D_{XY}^k$ during training. In other words, $\varepsilon$ serves not only to enhance expressiveness but also as a conditioning variable in $f_{\theta_Y}^{\text{aug-cond}}(y, \varepsilon)$. Consequently, during inference, if $\varepsilon_{n+1}$ correctly captures the source of the test sample, we can construct a smaller prediction set

$C_{\text{BNF}}^{\text{aug-cond}}(X_{n+1})$. Using a similar calculation as in Eq. (22), we derive

$$C_{\text{BNF}}^{\text{aug-cond}}(X_{n+1}) = \left\{ y : f_{\theta_Y}^{\text{aug-cond}}(y; \varepsilon_{n+1}) \in C_{\text{A}}(\overline{X}_{n+1}) \right\}, \text{ where } \overline{X}_{n+1} = f_{\theta_X}(X_{n+1}) \quad (68)$$

As shown in Figure 21, Augment-conditioned BNF leads to significantly smaller prediction sets compared to the standard Augmented BNF. Correia et al. (2024) also demonstrate that knowledge of the test sample's source can serve as valuable side information to improve prediction efficiency. Such side information is often available in real-world applications. For instance, in multi-center healthcare settings, models are required to generalize across different hospitals. In these scenarios, the center at which a patient is admitted is typically known and can be used to tailor the prediction procedure. Leveraging this information allows the model to generate tighter prediction sets while maintaining valid coverage guarantees.

We observe that the Augment-Conditioned BNF may exhibit slightly higher ASCG compared to the standard Augmented BNF. That is attributed to data sparsity, caused by the use of more distinct Gaussian distributions to model the data sources. As the number of Gaussians increases, the samples become more dispersed, making it more challenging to learn the underlying population distributions effectively. Hence, although the Augment-conditioned BNF yields smaller prediction sets, it compromises robustness in coverage, revealing an inherent trade-off.

## Q  LIMITATIONS

### Q.1  BEYOND MSDG: ALTERNATIVE FORMS OF JOINT DISTRIBUTION SHIFT

In Section 5, we describe the implementation of Augmented BNF within the context of multi-source domain generalization (MSDG), which represents a specific instance of joint distribution shift (Zou & Liu, 2024; Xu et al., 2025). Beyond MSDG, various alternative formulations have been proposed to characterize joint distribution shifts.

**Statistical distance ball.** The space of test distribution $Q_{XY}$ can be defined within a Wasserstein ball centered at the source distribution $P_{XY}$:

$$\mathcal{B}(P_{XY}, r) = \{Q_{XY} : W(P_{XY}, Q_{XY}) \le r\}, \quad \text{where } r \ge 0. \quad (69)$$

The notion of a Wasserstein ball can be generalized using alternative divergence measures, such as the Kullback–Leibler (KL) divergence (Cauchois et al., 2024).

**Input perturbation.** Joint distribution shift can be modeled as a perturbation applied to the test input $X_{n+1}$ (Gendler et al., 2021; Ghosh et al., 2023; Yan et al., 2024). In this formulation, the perturbed test input $\tilde{X}_{n+1}$ is constrained within a neighborhood of the original input by a norm-bound:

$$||\tilde{X}_{n+1} - X_{n+1}|| \le r, \quad \text{where } r \ge 0. \quad (70)$$

These two ways are closely related. For instance, sampling a collection of local pointwise perturbations around calibration points can approximate a global statistical distance ball. Thereby, we can replace the summation over $K$ domains in Eq. (26) with a supremum over a ball $\mathcal{B}(P_{XY}, r)$ by

$$\min_{\theta} [\sup_{Q_{XY} \in \mathcal{B}(P_{XY}, R)} W(P_{XY}, f_{\theta}^{\text{aug}} {}_{\#} Q_{XY})]. \quad (71)$$

Although the supremum over all possible distributions is theoretically well-defined, it is computationally intractable in practice; therefore, we approximate it following the two steps below:

(i) Initialize $Q_{XY}$ as emprical $P_{XY}$ samples

(ii) For fixed $\theta$, find a perturbed batch of samples that maximizes the objective Wasserstein distance but remains in $\mathcal{B}(P_{XY}, r)$.

This represents a sample-based estimation of the supremum, which can be used to update $\theta$. Further refinement is necessary to make it effective in practice. A key challenge lies in identifying a perturbed batch that maximizes the Wasserstein distance and remains within the specified ball. Several strategies can be employed to address this. For instance, genetic algorithms may be used to search for high-risk perturbations that maximize the divergence under the $f_{\theta}^{\text{aug}}$.

In this work, we do not propose a dedicated training algorithm for Augmented BNF to handle joint distribution shifts expressed in the two alternative forms discussed above. Developing such methods remains an important direction for future work.

## Q.2 ROOT-FINDING CHALLENGES

Unlike the original BNF, Augmented BNF does not employ a univariate monotonic transformation for the $Y$-branch. As a result, we cannot directly apply Eq. (18) to construct the prediction set. Instead, Augmented BNF relies on Eq. (22) to generate $C_{\text{BNF}}^{\text{aug}}(X_{n+1})$, which frames the construction as a root-finding problem. Specifically, let the interval endpoints of the calibrated set $C_{\text{A}}(\overline{X}_{n+1})$ be denoted by $y_{\text{lo}}$ and $y_{\text{hi}}$. Then, Eq. (22) requires solving a root-finding problem to identify the pre-images of these endpoints under the learned transformation. In particular, we need to find the values of $y$ that satisfy the following equations

$$f_{\theta_Y}^{\text{aug}}(y; \varepsilon_{n+1}) = y_{\text{lo}}; \quad f_{\theta_Y}^{\text{aug}}(y; \varepsilon_{n+1}) = y_{\text{hi}}. \tag{72}$$

While the coverage can be efficiently computed by checking whether $\overline{Y}_{n+1} \in C_{\text{A}}(\overline{X}_{n+1})$, as supported by Proposition 7, it is still crucial to develop a practical method for solving Eq. (72).

## Q.3 STOCHASTIC PREDICTION SETS

One practical drawback of Augmented BNF lies in its reliance on stochastic augmentation through a random noise $\varepsilon \sim \mathcal{N}(0, 1)$, which is used to modulate the $Y$-branch of the Augmented BNF in Eq. (21). While this augmentation introduces flexibility, it also introduces randomness into the transformation. As a result, in Eq. (22), the prediction set produced by Augmented BNF for the same input $x$ is no longer deterministic. The set varies across different forward passes depending on the realization of $\varepsilon$.

This stochasticity undermines one of the appealing features of standard conformal prediction: the deterministic and repeatable nature of the prediction set given a test point. In high-stakes domains, such randomness can lead to interpretability challenges or instability in downstream decisions. While one may average over multiple runs to approximate a stable prediction, this requires additional computational cost and still does not guarantee strict repeatability. This tradeoff between flexibility and determinism is a fundamental limitation when deploying Augmented BNF in sensitive applications.

## Q.4 ARCHITECTURAL INCOMPATIBILITY WITH ONE-DIMENSIONAL FEATURES

Another limitation of Augmented BNF stems from its architectural dependency on Real NVP (Dinh et al., 2016), a type of normalizing flow that is inherently designed for multi-dimensional transformations. Real NVP operates by alternating between dimensions of the input to apply affine coupling layers, as plotted in Figure 9. This necessitates a feature space $\mathcal{X}$ of at least two dimensions. Consequently, Augmented BNF inherits this constraint: its architecture presumes that the input feature $x$ is multivariate.

In the case where $x$ is one-dimensional, the affine coupling mechanism of Real NVP becomes undefined, rendering the model non-functional. As a result, Augmented BNF cannot be applied to tasks with univariate inputs. This presents a clear barrier for applying to domains, where no natural multivariate feature exists. One might consider artificially expanding $x$ with noise or engineered features to satisfy the dimensionality requirement, just like the augmented $Y$-branch in Eq. (21).

## Q.5 TUNING BIAS

In Section 4.2, the calibration set $\mathcal{S}_P$ participates in the training of the Augmented BNF, as described in Algorithm 1. However, this practice may undermine the rigor of conformal prediction, where calibration data is ideally held out from any training procedure. Despite this concern, similar strategies have been adopted in prior work (Angelopoulos et al., 2020; Dabah & Tirer, 2025; Xi et al., 2024; Yang & Kuchibhotla, 2024), often to simplify implementation. Notably, Zeng et al. (2025) identifies a parametric scaling law of tuning bias, showing that reusing calibration data introduces a bias that grows with model complexity and diminishes as the calibration set size increases.

To adhere more closely to the theoretical foundations of conformal prediction, a more principled approach would involve randomly partitioning $\mathcal{S}_P$ into two disjoint subsets: one used for training the Augmented BNF and another reserved exclusively for inference. Given the architectural complexity and parameterization of the Augmented BNF, as detailed in Appendix H, such a split is particularly

recommended to mitigate the risk of overfitting and maintain robust uncertainty guarantees. Nonetheless, in scenarios where calibration data is scarce, striking a balance between theoretical soundness and practical effectiveness remains an open challenge.

