# OpenReview forum: "Distributionally Robust Conditional Conformal Prediction"
_ICLR.cc/2026/Conference — Submitted to ICLR 2026_

### Official Review · Reviewer_vg7p · 2025-10-30

**Soundness:** 2
**Presentation:** 1
**Contribution:** 2
**Rating:** 2
**Confidence:** 4

**Summary:**

The paper studies constructing prediction intervals in the presence of distribution shifts from
the source to the target environment. The proposed method is based on the branched normalizing flow and mainly applies to the
case where the target environment is a mixture of source environments. The method is
tested in a rich set of numerical experiments.

**Strengths:**

1. The question studies in this paper is of interest, and the proposed method provides
some new perspectives on this problem.
2. The proposed method has been tested on a relatively rich sets of data.

**Weaknesses:**

1. The exposition of the manuscript could be improved. At times, the paper's motivation and contribution are misaligned. For example, it is claimed in the abstract that *"However, these methods often neglect the dependence between input features and conformal scores, leading to degraded conditional coverage for individual
test inputs"*, whereas the paper does not propose scores or methods to improve conditional coverage; instead, it focuses on mitigating conditional distribution shift. Elsewhere, some statements appear inaccurate—for instance, ‘these existing methods ignore the dependency of conformal scores on input features,’ which is not correct.

2. The proposed method requires learning a transformation form $P_{XY}$ to $Q_{XY}$,
which is in general impossible because $Y$ is not observed in the target environment and
therefore $Q_{Y\mid X}$ is not identifiable. The paper considers a special case
where $Q_{X,Y}$ is in the convex hull of the source distributions (where $Q_{Y \mid X}$ can be
identified from the shift of covariate distribution). This assumption feels restrictive;
it would help to discuss additional scenarios where the method is applicable or provide concrete examples illustrating its usefulness.


3. The guarantees provided by the model is a bit unclear. The results, in e.g. Appendix O,
leave terms like $\alpha_{\text{i.i.d.}}$ and $\alpha_{\text{trans}}$ without specifying how
these terms depend on the instances (e.g., sample sizes, dimensionality, number of sources, and shift magnitude).

**Questions:**

1. Continuing with my comment #2 in the "Weaknesses" section, I wonder if the authors could
provide other examples where the proposed method is applicable beyond the convex hull
model.

2. I am a bit confused by the bound in equation (14): is the right-hand side bound
really informative for the task at hand. To be specific, if the target is to evaluate
the conditional coverage, it is confusing to me why the covariate shift matters.
Consider the case of pure covariate shift, and we have a very accurate prediction model;
then one would expect the left-hand side to be very small while the right-hand side
to be large. In other words, minimizing the right-hand side is far from the true objective.

3. [1] also considers multi-source conformal prediction; how does the proposed method compare to theirs?

**Reference:**
1. Liu, Yi, et al. "Multi-source conformal inference under distribution shift." Proceedings of machine learning research 235 (2024): 31344.

---

> ### Author Response · Authors · 2025-11-23
>
> Thank you for your thoughtful and constructive comments. In direct response to your feedback, we uploaded a **new manuscript**. Our responses below will refer to this updated document and its new results to clarify how we have addressed each point.
>
> >For example, it is claimed in the abstract that "However, these methods often neglect the dependence between input features and conformal scores, leading to degraded conditional coverage for individual test inputs", whereas the paper does not propose scores or methods to improve conditional coverage; instead, it focuses on mitigating conditional distribution shift.
>
> In the abstract, our statement
>
> _“However, these methods often neglect the dependence between input features and conformal scores, leading to degraded conditional coverage for individual test inputs”_
>
> refers to a specific limitation of existing robustness methods: they typically align the marginal score distributions $P_V$ and $Q_V$ to maintain robust marginal coverage under shift, but do not enforce alignment of the conditional distributions $P_{V|x}$ and $Q_{V|x}$. When the conditional distributions vary with $x$, ignoring this dependence can lead to substantial conditional coverage errors at individual test inputs, even when marginal coverage is preserved.
>
> Our work does not introduce a new scoring function to directly improve conditional coverage in the i.i.d. setting. Rather, our contribution is to mitigate the mismatch between conditional score distributions that arises under distribution shift between the calibration and test data. We **revised the abstract** to more accurately reflect the paper’s technical focus.
>
> >Elsewhere, some statements appear inaccurate—for instance, ‘these existing methods ignore the dependency of conformal scores on input features,’ which is not correct.
>
> Thank you for pointing out the potential inaccuracy. Our intention was **not** to suggest that methods [1] and [2] assume that conformal scores are independent of input features. Rather, our point is that these approaches operate by aligning only the marginal score distributions  $P_V$ and $Q_V$ and do not model or correct the conditional distributions  $P_{V|x}$ and $Q_{V|x}$.
>
> Therefore, although they can improve marginal robustness, they do not address conditional distribution shift in the scores, which can lead to degraded conditional coverage for specific test inputs. We **revised the statement** to make this distinction clear.
>
> [1] Rina Foygel Barber, Emmanuel J Candes, Aaditya Ramdas, and Ryan J Tibshirani. Conformal
> prediction beyond exchangeability. The Annals of Statistics, 51(2):816–845, 2023.
>
> [2] Rui Xu, Chao Chen, Yue Sun, Parvathinathan Venkitasubramaniam, and Sihong Xie. Wasserstein-regularized conformal prediction under general distribution shift. In The Thirteenth International Conference on Learning Representations, 2025. URL https://openreview.net/forum?id=aJ3tiX1Tu4.
>
> >This assumption feels restrictive; it would help to discuss additional scenarios where the method is applicable or provide concrete examples illustrating its usefulness.
>
> The **experimental setup in Section 5** restricts the test distribution to be a random mixture of several elementary distributions. This choice is not an assumption required by our theory or method, but rather a **practical necessity** for controlled evaluation. Without structural assumptions on the shift, it is generally **intractable to learn or invert data transformations for an arbitrarily shifted distribution**. Existing related works typically need to constrain the distribution shift or request additional knowledge about it [3][4][5]. Multi-source domain generalization (MSDG) offers a widely studied and mathematically well-defined family, as introduced in **Section 2.2**.
>
> MSDG is not only standard but also **practically motivated**. **Appendix L.3** provides **four real-world examples** where natural distribution shifts arise precisely as a random mixture of several elementary distributions. These cases demonstrate that mixture-structured shifts occur organically in practice and are not artificially imposed. Thus, while our method leverages mixture information, it remains applicable to a meaningful and practically important class of distribution shifts commonly encountered in domain generalization research.
>
> [3] Ge Yan, Yaniv Romano, and Tsui-Wei Weng. Provably robust conformal prediction with improved efficiency. arXiv preprint arXiv:2404.19651, 2024.
>
> [4] Maxime Cauchois, Suyash Gupta, Alnur Ali, and John C Duchi. Robust validation: Confident predictions even when distributions shift. Journal of the American Statistical Association, pp.1–66, 2024
>
> [5] Subhankar Ghosh, Yuanjie Shi, Taha Belkhouja, Yan Yan, Jana Doppa, and Brian Jones. Probabilistically robust conformal prediction. In Uncertainty in Artificial Intelligence, pp. 681–690. PMLR,2023.

---

> ### Author Response · Authors · 2025-11-23
>
> >Continuing with my comment #2 in the "Weaknesses" section, I wonder if the authors could provide other examples where the proposed method is applicable beyond the convex hull model.
>
> Our approach is not limited to mixture shifts. As shown in **Appendix Q.1**, the method has potential for other forms of distribution shift, such as **perturbation-based shifts** or **shifts characterized by a statistical distance ball**, demonstrating that the technique extends beyond the mixture setting and is flexible in practice.
>
> >how these terms depend on the instances (e.g., sample sizes, dimensionality, number of sources, and shift magnitude).
>
> Regarding $\alpha_\text{trans}$, our empirical results highlight several key dependencies:
>
> (1). Number of sources: **Figure 17** shows that having more sources generally leads to a larger conditional coverage gap, indicating a higher $\alpha_\text{trans}$.
>
> (2). Sample size: **Figure 18** demonstrates that fewer calibration samples increase the coverage gap, also implying a larger $\alpha_\text{trans}$.
>
> (3). Dimensionality: As discussed in **Appendix G**, higher-dimensional input spaces exacerbate the curse of dimensionality, making it harder for normalizing flows to accurately learn the calibration and test distributions.
>
> (4). Shift magnitude: Interestingly, $\alpha_\text{trans}$ is not directly determined by standard metrics of distribution shift, such as Wasserstein distance or KL divergence. Instead, it is more related to the complexity of the calibration and test distributions. For example, a linear flow model can easily align the calibration and test distributions in **Figure 6**, yielding low $\alpha_\text{trans}$. However, as shown in **Figure 7**, when the distributions become more complex with real-world data, even widely used normalizing flow architectures, such as residual flows, may fail, resulting in a large ASCG.
>
> In summary, $\alpha_\text{trans}$ captures how challenging it is to align the calibration and test distributions given the number of sources, sample size, dimensionality, and distributional complexity, rather than simply the magnitude of the distribution shift.
>
> The magnitude of $\alpha_\text{i.i.d}$ primarily depends on the choice of the base conditional CP technique, such as conformalized quantile regression or localized conformal prediction. This aspect is outside the scope of our work, as our focus is on conditional coverage deviations arising specifically from distribution shift rather than the inherent limitations of the base conditional CP method.
>
> >[1] also considers multi-source conformal prediction; how does the proposed method compare to theirs?
>
> We are glad that the reviewer mentioned [3]. As noted, [3] focuses on achieving robust **marginal coverage** in the multi-source setup. Consequently, it does not need to consider conditional score distributions and only aligns the **marginal distributions**. Moreover, [3] assumes that the **marginal label distributions are identical** across sources and allows only covariate shifts. Under these assumptions, the empirical test feature data can be used to estimate the composition of the test distribution over sources, and the source weights can then be applied to compute a **weighted average of source quantiles**, achieving the desired marginal coverage on the test data.
>
> In contrast, our approach handles **both covariate and label shifts** and **does not infer how the test distribution is composed**. Instead, we leverage the generalization capability of normalizing flows to directly minimize the conditional coverage gap via Wasserstein optimal transport, enabling **robust conditional coverage**.
>
> [3] Liu, Yi, et al. "Multi-source conformal inference under distribution shift." Proceedings of machine learning research 235 (2024): 31344.

---

> ### Author Response · Authors · 2025-11-23
>
> >I am a bit confused by the bound in equation (14): is the right-hand side bound really informative for the task at hand. To be specific, if the target is to evaluate the conditional coverage, it is confusing to me why the covariate shift matters. Consider the case of pure covariate shift, and we have a very accurate prediction model; then one would expect the left-hand side to be very small while the right-hand side to be large. In other words, minimizing the right-hand side is far from the true objective.
>
> Indeed, while a large covariate shift can increase the Wasserstein distance $W(P_{XY},Q_{XY})$ on the right-hand side of Eq. (14), such shift may have limited impact on the conditional coverage gap. This apparent discrepancy is controlled by the parameter $η$ introduced in Eq. (14). As we further explain in **Appendix B, Lines 778–783**, $η$ quantifies how much of the discrepancy between $P_{XY}$ and $Q_{XY}$ is attributable to covariate shift versus concept shift.
>
> Intuitively, when $η$ is small, most of the distributional difference comes from covariate shift rather than changes in the conditional distributions $P_{Y|X}$ and $Q_{Y|X}$. In this case, the influence of $W(P_{XY},Q_{XY})$ on the coverage gap is limited, and the upper bound in Eq. (14) becomes tight. Thus, the bound is indeed informative: it properly reflects situations where covariate shift has little effect on conditional coverage.
>
> In our experiments, however, we do not explicitly estimate or incorporate $η$; instead, we minimize the Wasserstein distance between the joint distributions $P_{XY}$ and $Q_{XY}$. This may appear conservative (“overkill”), but it is practically necessary. Empirically minimizing the conditional Wasserstein distance $W(P_{Y|X}, Q_{Y|X})$ is extremely challenging with finite samples, especially in moderate or high dimensions. Moreover, the scenario *"a very accurate prediction model"* described in the reviewer’s comment is rarely attainable in practice. Existing conditional conformal methods (e.g., conformalized quantile regression, as discussed in **Appendix I**, and localized conformal prediction) are highly sensitive to covariate shift in the sampled data and can suffer substantial degradation in conditional coverage even under moderate shift.
>
> Therefore, while covariate shift alone may not theoretically enlarge the ICG, minimizing the full Wasserstein distance serves as a robust and practically reliable surrogate, ensuring that both covariate and conditional discrepancies are jointly controlled. This robustness is crucial because real-world conditional CP models seldom achieve perfect accuracy or stability under covariate shift.

---

### Official Review · Reviewer_ukCp · 2025-10-31

**Soundness:** 3
**Presentation:** 1
**Contribution:** 2
**Rating:** 4
**Confidence:** 3

**Summary:**

This paper addresses conditional coverage under calibration-test distribution shift. It (i) defines a CCG and its expectation over the test feature law ICG as measures of conditional coverage robustness under distributional shift; (ii) derives upper bounds on ICG using Wasserstein distance between calibration and test joint distributions. Building on this insight, the authors propose Branched Normalizing Flow (BNF) and an Augmented BNF variant which attempt to map the test joint distribution to the calibration joint via an invertible flow. After transforming inputs, they apply an adaptive conformal algorithm on the normalized data and inverse-transform prediction sets to the original label space. Empirically, they evaluate on synthetic and real datasets and introduce ASCG as an empirical metric for conditional coverage. The results show improved ASCG and stable marginal coverage compared to the baseline methods.

**Strengths:**

1.	Figure 1 is very detailed and helps in understanding the presented framework.
2.	The empirical results show that the proposed method performs better than the baseline methods.

**Weaknesses:**

1.	The abstract and introduction are somewhat misleading: They describe a general distribution shift between the calibration and test, while the proposed method requires additional information about the shift.
2.	Limited applicability: the proposed approach requires knowledge about the test distribution, unlike many existing methods that consider a general distribution shift.
3.	The theoretical assumptions are hard to verify in practice.
4.	The metric ASCG only considers marginal coordinates and ignores multivariate conditional failures. Also, the effect of M is not analyzed. Furthermore, I believe the authors should compare the methods using the WSC metric as well.

**Questions:**

### Questions
1. In line 212, the paper suggests limiting the sensitivity of the score to changes in y conditioned on x. However, shouldn’t we want the score function to remain responsive to variations in y given x? Could the authors clarify this point?
### Minor comments
1. The worst slab coverage was first introduced in “Knowing what You Know: valid and validated confidence sets in multiclass and multilabel prediction” by Cauchois et al.

---

> ### Author Response · Authors · 2025-11-23
>
> Thank you for your thoughtful and constructive comments. In direct response to your feedback, we uploaded a **new manuscript**. Our responses below will refer to this updated document and its new results to clarify how we have addressed each point.
>
> >The metric ASCG only considers marginal coordinates and ignores multivariate conditional failures.
>
> We address this concern by extending ASCG to a multivariate setting following the slab-based construction used in WSC. Specifically, as detailed in **Appendix M.3**, we generate random multivariate slabs to induce partitions over higher-dimensional subspaces. This allows the metric to capture multivariate conditional failures beyond one-dimensional slices. As demonstrated in **Figure 16**, the resulting multivariate ASCG remains low, indicating that our method preserves conditional coverage even under multivariate partitions of the feature space.
>
> > Also, the effect of M is not analyzed
>
> The choice of the number of slices $M$ affects the sensitivity of ASCG. As $M$ decreases, each slice $\mathcal{S}_{i,m}$ defined in Eq.(27) becomes larger, and when $M=1$ each slice covers the entire test set $\mathcal{S}_Q$; in this extreme, ASCG in Eq.(28) reduces to the marginal coverage gap, which cannot detect localized conditional failures. Conversely, as $M$ becomes very large, slices become very small, making ASCG highly sensitive to sampling noise and possibly overestimating coverage deviations. Therefore, there is a practical trade-off in choosing $M$: it should be large enough to reveal meaningful localized invalid coverage but not so large as to make the statistic dominated by small-sample variability.
>
> > Furthermore, I believe the authors should compare the methods using the WSC metric as well.
>
> Thank you for this suggestion. We agree that reporting worst-slice coverage (WSC) is important for comparison with prior work.
>
> As requested, we have included the WSC results for all methods in **Figure 13** of **Appendix M.1**. The data shows that our method consistently maintains WSC values near the target coverage level of 0.9.
>
> We do observe that the Worst-Case Conformal Prediction (WC-CP) baseline also achieves high WSC; however, this result is primarily attributable to its overly conservative prediction sets, which lead to significant over-coverage across most slices (as visualized in **Figure 10**). Because the WSC metric defined in Eq. (54) only considers the infimum of coverage, it does not penalize this type of over-coverage.
>
> This insight that a method can perform well on the WSC metric by being universally conservative was a key motivation for our use of the average-slice coverage gap (ASCG). We believe ASCG provides a more balanced and informative evaluation by quantifying deviation from the target coverage across all slices, thereby penalizing both under-coverage and over-coverage.
>
> Moreover, in **Appendix M.2**, we propose worst-slice coverage gap (WSCG) by taking the maximum absolute difference from the desired confidence level $1-\alpha$. Our approach consistently achieves low WSCG values across datasets and confidence levels in **Figure 14** and **Figure 15**.

---

> ### Author Response · Authors · 2025-11-23
>
> > The abstract and introduction are somewhat misleading: They describe a general distribution shift between the calibration and test, while the proposed method requires additional information about the shift. Limited applicability: the proposed approach requires knowledge about the test distribution, unlike many existing methods that consider a general distribution shift.
>
> The reviewer’s concern arises from the experimental setup in Section 5, where we restrict the test distribution to be a random mixture of several elementary distributions. This choice is not an assumption required by our theory or method, but rather a **practical necessity** for controlled evaluation. Without structural assumptions on the shift, it is generally **intractable to learn or invert data transformations for an arbitrarily shifted distribution**. Existing related works typically need to constrain the distribution shift or request additional knowledge about it [1][2][3]. Multi-source domain generalization (MSDG) offers a widely studied and mathematically well-defined family, as introduced in **Section 2.2**.
>
> MSDG is not only standard but also **practically motivated**. **Appendix L.3** provides four real-world examples where natural distribution shifts arise precisely as a random mixture of several elementary distributions. These cases demonstrate that mixture-structured shifts occur organically in practice and are not artificially imposed. Thus, while our method leverages mixture information, it remains applicable to a meaningful and practically important class of distribution shifts commonly encountered in domain generalization research.
>
> Importantly, our approach is not limited to mixture shifts. As shown in **Appendix Q.1**, the method has potential for other forms of distribution shift, such as **perturbation-based shifts** or **shifts characterized by a statistical distance ball**, demonstrating that the technique extends beyond the mixture setting and is flexible in practice.
>
> [1] Rui Xu, Chao Chen, Yue Sun, Parvathinathan Venkitasubramaniam, and Sihong Xie. Wasserstein-regularized conformal prediction under general distribution shift. In The Thirteenth International Conference on Learning Representations, 2025.
>
> [2] Ge Yan, Yaniv Romano, and Tsui-Wei Weng. Provably robust conformal prediction with improved efficiency. arXiv preprint arXiv:2404.19651, 2024.
>
> [3] Maxime Cauchois, Suyash Gupta, Alnur Ali, and John C Duchi. Robust validation: Confident predictions even when distributions shift. Journal of the American Statistical Association, pp.1–66, 2024
>
> > In line 212, the paper suggests limiting the sensitivity of the score to changes in y conditioned on x. However, shouldn’t we want the score function to remain responsive to variations in y given x? Could the authors clarify this point?
>
> While it may seem desirable for the score function to be fully sensitive to variations in $y$ given $x$, excessive sensitivity is actually detrimental to the robustness of conformal prediction, especially under distribution shift or in the presence of noise. Recent work (e.g., Lipschitz-bounded conformal scores [4]) has shown that constraining the score’s sensitivity helps stabilize the conformal procedure.
>
> [4] Massena, Thomas, et al. Efficient robust conformal prediction via Lipschitz-bounded networks. arXiv preprint arXiv:2506.05434 (2025).
>
> >The worst slab coverage was first introduced in “Knowing what You Know: valid and validated confidence sets in multiclass and multilabel prediction” by Cauchois et al.
>
> Thank you for pointing this out. We have modified the corresponding citations.
>
> >The theoretical assumptions are hard to verify in practice.
>
> We acknowledge that the theoretical assumptions, such as **Eq. (11) in Theorem 2**, may appear difficult to verify directly in practical settings. Their purpose, however, is to characterize the structural factors that fundamentally govern conditional coverage under distribution shift.
>
> For example, Eq. (11) formalizes a key requirement: the conditional discrepancy $W(\mu_{Y|x},\nu_{Y|x})$ should remain controlled by the overall joint shift $W(\mu_{XY},\nu_{XY})$.  Eq. (11) serves as a conceptual guide for designing algorithms that adapt to the geometry of the joint shift. In practice, their implications are reflected in our empirical findings: across diverse test distributions (**Figure 3**), our method consistently maintains reliable conditional coverage, indicating that the structural regimes captured by the theory are realistic for real-world shifts.

---

### Official Review · Reviewer_9Bgi · 2025-10-31

**Soundness:** 3
**Presentation:** 1
**Contribution:** 3
**Rating:** 4
**Confidence:** 3

**Summary:**

Suppose we have procedure that generates prediction sets $C$ such that $P_{(X,Y) \sim P_{XY}} (Y \in C(X) \mid X=x) = 1-\alpha$ for all $x$ (CQR, for example, is one way to approximately achieve this). In general, $C$ will no longer have this $1-\alpha$ conditional coverage if instead $(X,Y) \sim Q_{X,Y}$. Now suppose we have invertible mappings $f_X$ and $f_Y$ such that for $(X,Y) \sim Q_{XY}$, we have $(f_X(X), f_Y(Y)) \sim P_{XY}$. Then, for $(X,Y) \sim Q_{XY}$, we can construct the conditional-coverage set in the mapped space and then map back to the original space, inheriting the conditional coverage property from the mapped space.

Theoretically, the paper provides an upper bound on the (integrated) conditional-coverage gap between $P_{X,Y}$ and $Q_{X,Y}$ in terms of the Wasserstein distance between $P_{XY}$ and $Q_{XY}$. This is useful in the multi-source domain generalization setting, as it suggests an optimization objective for learning the mapping (normalizing flow) that is in terms of the Wasserstein distance. The method is validated on several synthetic and real world datasets and the proposed method is shown to decrease the average-slice coverage gap.

**Strengths:**

The problem the paper solves of maintaining conditional-coverage under distribution shift is an interesting one, and the proposed solution is also conceptually interesting.

**Weaknesses:**

The writing is severely lacking and does not do justice to the underlying ideas, which are nice but currently presented in an inaccessible way. This is true at a micro level (typos, grammar mistakes, and weird/ambiguous wording) and also at a macro level (the information is not presented in an optimal way). I will elaborate on a few places for improvement, but not comprehensively. I encourage the authors to spend time checking grammar and thinking about ways to further simplify the presentation.

First, as context: I am quite familiar with conformal prediction but have very limited knowledge of distributional robustness. Ideally, your paper should be accessible to people like me.

* Line 40: depend on *the* specific test input
* Line 49-50: “Various upper bounds .. proposed to maintain marginal coverage guarantee” — what do you mean by this? I think perhaps it is not so much “maintain” but rather to derive $\epsilon$ in $P(Y \in C(X)) \geq 1-\alpha-\epsilon$
* Line 85-94: As someone with no distributional robustness background, I got nothing from these paragraphs. Is there a way to briefly provide either more of the DR background that is needed or to describe this in a more intuitive way?
* Line 95: I don’t know what “To enhance expressiveness” means here
* Line 97: “sale prediction” -> “sales prediction”?
* Line 100: The results *show*
* Line 116: independent *from*
* Line 119: “Theoretically” -> “Formally”
* Line 125: “almost inaccessible” -> not practically achievable
* Line 164: The double subscript in $F_{P_{V|x}}$ is a lot. Can you just write $F_P$?
* Definition 1: I would consider moving this, or something like it, to the introduction. If you want to talk about Wasserstein distance in the introduction, it should be defined (as least informally)
* Line 199: 2-product *metric*
* Eq 15: I spent a while trying to figure out what $\bar{X}_{n+1}$ and $\bar{Y}_{n+1}$ had been defined, only to realize that you were defining the output of the mapping to be that. I would reorder this and write $(\bar{X}_{n+1}, \bar{Y}_{n+1}) := f_(\theta)(…)$
* Line 255-257: These requirements on $f_{\theta}$ are not very clear. Can you also write it out mathematically?
* Line 286: *has* the conditional guarantee
* Line 304: There is an extra space before the first word
* Sec 4.2: What is the intuition for why the Gaussian noise helps?
* Line 342: You should mention that this is the expectation assuming $\lambda$ is uniform on the simplex
* Line 345+: I believe “calibration set” is meant throughout and not “training set”

My overall comment about the Introduction is that there is too much technical content/jargon that should be explained more carefully if there is space; otherwise, it would be better to provide more high-level/intuitive summaries. I also found Figure 1 quite hard to follow.

**Questions:**

* How do you ensure invertibility of the normalizing flow? More background on distributional robustness would be useful throughout the paper.
* It is fine that you use average-slice coverage gap as your main metric, but I would still like to see the results for worst-slice coverage, as this is the metric used in previous papers. Do you have these results in the Appendix somewhere?
* The theory is described in the case where we have a calibration distribution $P_{XY}$ and a single test distribution $Q_{XY}$. Does your method apply to this setting? How does it perform?

Overall, I think the paper has a lot of promise and I would be happy to increase my score if these weaknesses can be addressed, especially the writing quality.

---

> ### Author Response · Authors · 2025-11-23
>
> Thank you for your thoughtful and constructive comments. In direct response to your feedback, we uploaded a **new manuscript**. Our responses below will refer to this updated document and its new results to clarify how we have addressed each point.
>
> >How do you ensure invertibility of the normalizing flow? More background on distributional robustness would be useful throughout the paper.
>
> To clarify, invertibility is guaranteed by construction in normalizing flows. A normalizing flow, denoted by $f$, is explicitly designed to be bijective with a tractable Jacobian determinant. As a result, the overall transformation $f$ is invertible, and its inverse $g=f^{-1}$ is computable.
>
> For completeness, let $q_{XY}$ denote the density of the distribution $Q_{XY}$. Given a bijective flow $f$,  Let $f_{\\#}Q_{XY}$ denote the pushforward distribution of $Q_{XY}$ under the bijective transformation $f$, whose density is written as $f_{\\#} q_{XY}$.
> According to the change-of-variables formula, we can derive
> $f_{\\#} q_{XY}\left(f(x,y)\right)=q_{XY}(x,y)\left|\det \frac{\partial f(x,y)}{\partial (x,y)}\right|^{-1},$
> ensuring exact density computation.
>
> The parameters of flow $f$ are learned by aligning the transformed density $f_{\\#} q_{XY}$ with the reference calibration distribution density $p_{XY}$.
> This procedure is a standard approach in distributional robustness: we learn an invertible transformation that maps the shifted distribution $Q_{XY}$ to a reference distribution $P_{XY}$, while retaining the inverse map for use at test time.
>
> >It is fine that you use average-slice coverage gap as your main metric, but I would still like to see the results for worst-slice coverage, as this is the metric used in previous papers. Do you have these results in the Appendix somewhere?
>
> Thank you for this suggestion. We agree that reporting worst-slice coverage (WSC) is important for comparison with prior work.
>
> As requested, we have included the WSC results for all methods in **Figure 13** of **Appendix M.1**. The result shows that our method consistently maintains WSC values near the target coverage level of 0.9.
>
> We do observe that the Worst-Case Conformal Prediction (WC-CP) baseline also achieves high WSC; however, this result is primarily attributable to its overly conservative prediction sets, which lead to significant over-coverage across most slices (as visualized in **Figure 10**). Because the WSC metric defined in Eq. (54) only considers the infimum of coverage, it does not penalize this type of over-coverage.
>
> This insight that a method can appear to perform well under the WSC metric merely by being universally conservative was a key motivation for our use of the Average-Slice Coverage Gap (ASCG). We believe ASCG provides a more balanced and informative evaluation by quantifying deviation from the target coverage across all slices, thereby penalizing both under-coverage and over-coverage.
>
> Moreover, in **Appendix M.2**, we propose worst-slice coverage gap (WSCG) by taking the maximum absolute difference from the desired confidence level $1-\alpha$. Our approach consistently achieves low WSCG values across datasets and confidence levels in **Figure 14** and **Figure 15**.
>
> >The theory is described in the case where we have a calibration distribution  and a single test distribution . Does your method apply to this setting? How does it perform?
>
> Our method **directly applies** to the setting with one calibration distribution and one unknown test distribution. In fact, this is exactly the scenario considered in our theoretical analysis: the test distribution may be any (unknown) mixture of the elementary source distributions, and our goal is to achieve reliable conditional coverage under this single mixture distribution.
>
> In the experiments, however, we evaluate performance over multiple randomly sampled test mixtures. This is not because our method requires multiple test distributions, but rather to *demonstrate robustness across all possible mixture compositions*. Each sampled mixture corresponds to one specific test distribution, and for each of them, our method is evaluated independently.
>
> To make this clear and statistically meaningful, we report the results using **box plots in Figure 3**, instead of showing only mean values. This presentation highlights that:
>
> (1) **For every single test mixture distribution**, our method achieves consistently low conditional coverage errors, and
>
> (2) **The performance variability across mixtures is small**, indicating stability with respect to how the test distribution is composed.
>
> Thus, the experimental design provides strong empirical evidence that our method performs well not just on average, but **for each individual test distribution in the single-distribution setting**.

---

> ### Author Response · Authors · 2025-11-23
>
> > The writing is severely lacking and does not do justice to the underlying ideas, which are nice but currently presented in an inaccessible way. This is true at a micro level (typos, grammar mistakes, and weird/ambiguous wording) and also at a macro level (the information is not presented in an optimal way). I will elaborate on a few places for improvement, but not comprehensively. I encourage the authors to spend time checking grammar and thinking about ways to further simplify the presentation.
>
> We thank the reviewers for their valuable and constructive suggestions. In this **updated document**, we have **prioritized addressing the key technical concerns and substantive experimental requests** to solidify our core technical contributions, while also **incorporating the majority of the suggested writing improvements**.
>
> For the more extensive revisions, such as a comprehensive rewrite of the introduction and a redesign of Figure 1, we plan to undertake these important structural improvements for a subsequent version of the manuscript.
>
> > Sec 4.2: What is the intuition for why the Gaussian noise helps?
>
> Gaussian noise in Augmented Normalizing Flows helps because it effectively expands the space in which the bijective mapping operates, giving the model more geometric freedom while still remaining invertible. Although the flow remains a bijection in the augmented space, the added noise dimensions act like latent variables that absorb structural complexity that would otherwise force the deterministic mapping to twist or warp unnaturally. This extra “breathing room” allows the flow to represent multimodal or highly curved distributions more smoothly, without extreme Jacobian distortions, and results in a much better-conditioned optimization landscape.
>
> Intuitively, the noise lifts the data into a higher-dimensional space where complicated structures become easier to model, making the behavior of the flow closer to that of a flexible latent-variable model while preserving exact likelihoods and invertibility. We refer to **Figure 1 in [1]** as a very illustrative example.
>
> [1] Huang C W, Dinh L, Courville A. Augmented normalizing flows: Bridging the gap between generative flows and latent variable models[J]. arXiv preprint arXiv:2002.07101, 2020.
>
> > Line 255-257: These requirements on $f_\theta$  are not very clear. Can you also write it out mathematically?
>
> We thank the reviewer for this request for clarification. The requirements for the transformation $f_\theta$ are indeed critical and can be stated more precisely as follows:
>
> **Requirement (i): Invertibility Preserving Prediction Intervals.**
>
> Let $C_A (\bar{X} _{n+1})=[\bar{l} _\text{low},\bar{l} _\text{high}]$ be the prediction interval produced in the transformed space for the transformed input $\bar{X} _{n+1}$.
>
> To ensure valid predictions in the original space, the inverse transform must recover the corresponding interval endpoints for the same input $X_{n+1}$:
>
> $(X _{n+1}, l _\text{low} ):=f _\theta^{-1} (\bar{X} _{n+1},\bar{l} _\text{low})$,
>
> $(X _{n+1}, l _\text{high}):=f _\theta^{-1} (\bar{X} _{n+1},\bar{l} _\text{high})$.
>
> Thus, the prediction set in the original space is the pullback: $C_A (X _{n+1})=[l _\text{low},l _\text{high}]$.
>
> The invertibility of $f_\theta$ must allow us to convert the latent-space interval endpoints back to the original label-space endpoints while keeping the same input $X_{n+1}$. This ensures that conditional validity in the transformed space transfers back to the original space.
>
> **Requirement (ii): The transform must be computable without access to $Y_{n+1}$**
>
> At test time, $Y_{n+1}$is unknown. Therefore, the transformation must factor as:
> $f_θ (x,y)=(f_{θ_X} (x),f_{θ_Y} (y))$ for some deterministic mapping $\bar{X}=f_{θ_X} (X)$. This factorization ensures that $\bar{X} _{n+1}$ can be computed from ${X} _{n+1}$ alone, without knowledge of $Y _{n+1}$, making the procedure feasible for prediction.

---

### Official Review · Reviewer_8TcN · 2025-11-01

**Soundness:** 2
**Presentation:** 2
**Contribution:** 2
**Rating:** 2
**Confidence:** 3

**Summary:**

This paper proposes a distributionally robust framework for conditional conformal prediction to maintain valid conditional coverage guarantees under distribution shifts. The authors first introduce the Conditional Coverage Gap (CCG) to quantify coverage robustness at individual test points and its expectation, the Integrated Coverage Gap (ICG), as an overall robustness measure. Theoretically, they derive an upper bound for ICG using the Wasserstein distance between the calibration and test joint distributions, revealing how data-level shifts propagate to the conformal score space. Motivated by this bound, they design a Branched Normalizing Flow (BNF) that learns an invertible mapping between the test and calibration distributions via Wasserstein minimization. Its branched structure enables separate transformation of inputs and labels, allowing test-time input normalization without the true label. An Augmented BNF variant enhances expressiveness using Gaussian noise.

**Strengths:**

It introduces a theoretically grounded framework for achieving distributionally robust conditional coverage in conformal prediction, formalized through new metrics—the Conditional Coverage Gap (CCG) and Integrated Coverage Gap (ICG)—and linked to the Wasserstein distance between distributions.

**Weaknesses:**

The paper appears to mix many concepts including adaptivity of prediction sets, conditional coverage, and the shift of conditional distributions. While these elements may share some qualities, they are fundamentally different and often can be solved in distinct ways. For example, the motivation proposing IGG is $F_{P_{V|x}}(\tau(x))$ is different from $F_{Q_{V|x}}(\tau(x))$ when $P_{V|x} \neq Q_{V|x}$, but even if there is no gap between $F_{P_{V|x}}(\tau(x))$and $F_{Q_{V|x}}(\tau(x))$, it doesn’t mean that the conditional coverage is guaranteed. And even if $P_{V|x} = Q_{V|x}$, there is also not conditional coverage guarantee. Besides , there is a problem in mathematics that $\tau(x)$is not only depended on $X_{n+1} = x$, but depended on the calibration data, so the probability of (3) takes over the product measure of test data and calibration data. In this case, $P(s(X_{n+1}, Y_{n+1} \leq \tau (x))|X_{n+1} = x)$ $\neq F_{Q_{V|x}}(\tau(x))$ and $P(s(X_{n+1}, Y_{n+1} \leq \tau (x))|X_{n+1} = x)$ $\neq F_{P_{V|x}}(\tau(x))$.

**Questions:**

1. About  Augmented BNF, how does $\varepsilon$makes an effect on $f_{\theta_{Y}}^{aug}(y;\varepsilon)$?
2. How does Theorem3 suggest a surrogate objective for Augmented BNF by $\sum_{k=1}^{K} \lambda_{k}W(P_{XY}, f_{\theta}^{aug},$ #$D^{k}_{XY}).$
3. Why Algorithm 1 loop N epochs?
4. please supplement the detailed experimental settings such as the form of $f_{\theta_Y}(x, \varepsilon)$ and your datas' distribution.
5. What is the relationship between CCG and conditional conformal prediction? Why do you bound it?

---

> ### Author Response · Authors · 2025-11-23
>
> Thank you for your thoughtful and constructive comments. In direct response to your feedback, we uploaded a **new manuscript**. Our responses below will refer to this updated document and its new results to clarify how we have addressed each point.
>
> >The paper appears to mix many concepts including adaptivity of prediction sets, conditional coverage, and the shift of conditional distributions. While these elements may share some qualities, they are fundamentally different and often can be solved in distinct ways. For example, the motivation proposing IGG is $F_{P_{V|x}}(\tau(x))$ is different from $F_{Q_{V|x}}(\tau(x))$ when $P_{V|x} \neq Q_{V|x}$, but even if there is no gap between $F_{P_{V|x}}(\tau(x))$and $F_{Q_{V|x}}(\tau(x))$, it doesn’t mean that the conditional coverage is guaranteed. And even if $P_{V|x} = Q_{V|x}$, there is also not conditional coverage guarantee
>
> We think the reviewer may be mixing up two related but distinct aspects of our work:
>
> (1). Theoretical analysis (Eqs. (1) and (2)): In **Lines 119–123**, we consider an idealized setting where we have **enough calibration samples with** $X_i=x$. In this case, and when there is no distribution shift ($P_{V|x}=Q_{V|x}$), conformal prediction (CP) indeed achieves conditional coverage at least $1-\alpha$. This part of our work is purely theoretical, focusing on the regime where the quantile is computed from the exact conditional scores $\\{V_i: X_i = x\\}_{i=1}^n$.
>
> (2). Practical implementation: In **Lines 125–135**, we discuss what to do when calibration samples at the exact test input $x$ are **limited or unavailable**. Existing methods like nearest neighbors and quantile regression address this practical challenge. They are not the theoretical property studied in Eqs. (1)–(2).
>
> In short, when thinking about conditional coverage under distribution shift, it helps to separate **distribution shift issues**, which our theory focuses on, and **practical limitations** due to the scarcity of calibration points at $x$.
>
> Moreover, to clearly distinguish these two aspects, we introduce two quantities, $\alpha_\text{trans}$ and $\alpha_\text{i.i.d}$, in **Appendix O.2**, corresponding to aspects (1) and (2), respectively.
>
> We hope this clarifies that our manuscript treats these two challenges separately and consistently.
>
> > Besides , there is a problem in mathematics that $\tau(x)$ is not only depended on $x$, but depended on the calibration data, so the probability of (3) takes over the product measure of test data and calibration data. In this case, $\text{Pr}\left(s(X_{n+1},Y_{n+1}\leq \tau (x))|X_{n+1}=x\right) \neq F_{Q_{V|x}}(\tau(x)) $ and $\text{Pr}\left(s(X_{n+1},Y_{n+1}\leq \tau (x))|X_{n+1}=x\right) \neq F_{P_{V|x}}(\tau(x)) $.
>
> We admit the quantile $\tau$ depends both on calibration data and test input $x$. A more rigorous way to express the quantile should be $\tau (x,\\{V_i: X_i = x\\}_{i=1}^n) = \text{Quantile}(⌈(1 − α)(n_x + 1)⌉/n_x, \\{V_i: X_i = x\\} _{i=1}^n)$ where $n_x$ is the number of calibration samples whose feature is $x$.
>
> We assume that in your comment, $s(X _{n+1},Y _{n+1}\leq\tau (x))$ should be interpreted as $s(X _{n+1},Y _{n+1})\leq\tau(x,\\{V _i: X _i = x\\} _{i=1}^n)$.
>
> By definition, when $X_{n+1}=x$, the test conformal score $s(x, Y_{n+1})$ follows the conditional distribution $Q _{V|x}$. Moreover, $\tau (x,\\{V _i: X _i = x\\} _{i=1}^n)$ is defined as a quantile value of $P _{V|x}$. When there is no distribution shift ($P _{V|x}=Q _{V|x}$), evaluating the CDFs of $P _{V|x}$ and $Q _{V|x}$ at this quantile yields
>
> $\text{Pr}\left(s(x,Y _{n+1})\leq \tau (x,\\{V _i: X _i = x\\} _{i=1}^n) \right)=F _{Q _{V|x}}\left(\tau (x,\\{V _i: X _i = x\\} _{i=1}^n)\right)= F _{P _{V|x}}\left(\tau (x,\\{V _i: X _i = x\\} _{i=1}^n)\right)$,
>
> which follows directly from the definition of the conditional distribution and its CDF evaluated at a quantile.

---

> ### Author Response · Authors · 2025-11-23
>
> >About Augmented BNF, how does $\epsilon$ make an effect
>
> Gaussian noise in Augmented Normalizing Flows helps because it effectively expands the space in which the bijective mapping operates, giving the model more geometric freedom while still remaining invertible. Although the flow remains a bijection in the augmented space, the added noise dimensions act like latent variables that absorb structural complexity that would otherwise force the deterministic mapping to twist or warp unnaturally. This extra “breathing room” allows the flow to represent multimodal or highly curved distributions more smoothly, without extreme Jacobian distortions, and results in a much better-conditioned optimization landscape.
>
> Intuitively, the noise lifts the data into a higher-dimensional space where complicated structures become easier to model, making the behavior of the flow closer to that of a flexible latent-variable model while preserving exact likelihoods and invertibility. We refer to **Figure 1 in [1]** as a very illustrative example.
>
> [1] Huang C W, Dinh L, Courville A. Augmented normalizing flows: Bridging the gap between generative flows and latent variable models[J]. arXiv preprint arXiv:2002.07101, 2020.
>
> > How does Theorem3 suggest a surrogate objective for Augmented BNF by $\sum _{k=1}^K \lambda _kW(P _{XY},f ^{\text{aug}} _{\theta\\#}D ^k _{XY})$
>
> **Eq. (25)** in Theorem 3 shows that the Wasserstein distance to a mixture distribution can be upper bounded by the mixture of Wasserstein distances to the individual components. Noting that
> $f ^{\text{aug}} _{\theta\\#}Q _{XY}$ is a mixture of $\\{f ^{\text{aug}} _{\theta\\#}D ^k _{XY}\\} _{k=1}^K$, we can therefore upper bound the Wasserstein distance $W(P _{XY},f ^{\text{aug}} _{\theta\\#}Q _{XY})$ by $\sum _{k=1}^K \lambda _kW(P _{XY},f ^{\text{aug}} _{\theta\\#}D ^k _{XY})$ based on Eq. (25), which naturally serves as a surrogate objective for optimization.
>
> >Why Algorithm 1 loop N epochs?
>
> The normalizing flow requires training and we train it for $N$ epochs.
>
> >please supplement the detailed experimental settings such as the form of $f_{\theta_Y}(x,\epsilon)$ and your datas' distribution.
>
> The detailed architecture of the BNF, including its formulation and components, is specified in **Appendix H** and illustrated in **Figure 9**. The distribution of our datasets is visualized in **Figure 12** of **Appendix L**. To visualize potential distribution shifts in our high-dimensional data, we employed t-SNE to project the data into a 2D space. The visualization confirms clear distributional shifts between most data sources. We note that in some specific cases, such as between the second and third sources in the Bike Rental and Fair Med setups, the distributions exhibit some similarity. We would like to clarify that this overlap is not an artifact of our experimental design but a natural characteristic of the real-world datasets we used, reflecting genuine similarities in the underlying scenarios from which the data was collected.
>
> >What is the relationship between CCG and conditional conformal prediction? Why do you bound it?
>
> Conditional conformal prediction is a class of methods (e.g., conformalized quantile regression, localized conformal prediction) designed to approximate valid conditional coverage as formalized in Eq. (2) under the **i.i.d. setting**. The Conditional Coverage Gap (CCG), in contrast, is a specific theoretical metric we use to quantify the invalidity that arises specifically from **distribution shift** between the calibration and test distributions, separate from the finite-sample error caused by a limited number of calibration points. We bound the CCG precisely to analytically dissect how and to what extent a distribution shift leads to a breakdown in conditional coverage.
>
> We acknowledge that in practice, the finite size of the calibration set makes the ideal conditional guarantee in Eq. (2) statistically unattainable. However, it is meaningful to analyze the theoretical regime with a sufficient number of samples at $X_i=x$ to isolate and study the impact of distribution shift on conditional coverage. This allows us to cleanly separate the error introduced by distribution shift from the error introduced by finite-sample estimation.

---

> ### Comment · Reviewer_8TcN · 2025-11-24
>
> The most crucial issue in your paper is that you don't illustrate IGG or CGG how direct influence conditional converage loss i.e., there should be a theorem in the form of $\mathbb{P}(Y_{n+1} \in C(X_{n+1})) \geq 1 - \alpha - f(IGG)$. Otherwise, the paper just proposes an abstract definition and bound it with Wasserstein distance, which doesn't have no practical meaning. An excellent similar example is [1].
>
> [1] Barber, R. F., & Pananjady, A. (2025). Predictive inference for time series: why is split conformal effective despite temporal dependence?. arXiv preprint arXiv:2510.02471.

---

> ### Author Response · Authors · 2025-11-24
>
> We thank the reviewer for the suggestion and have rewritten our analysis explicitly in the recommended form.
>
> Under distribution shift, where the test sample $(X_{n+1},Y_{n+1})\sim Q_{XY}$ differs from the calibration distribution $P_{XY}$, we define the **conditional coverage gap (CCG)** in Eq.(4) as
>
> $\text{CCG}(P,Q,x)= \left|F_{P_{V|x}}\left(\tau(x,\\{V _i:X _i=x\\} _{i=1} ^n)\right)-F _{Q _{V|x}}\left(\tau(x,\\{V _i:X _i=x\\} _{i=1} ^n)\right)\right|$,
>
> which implies
>
> $F _{Q _{V|x}}\left(\tau(x,\\{V _i:X _i=x\\} _{i=1} ^n)\right)\geq F _{P _{V|x}}\left(\tau(x,\\{V _i:X _i=x\\} _{i=1} ^n)\right)-\text{CCG}(P,Q,x)$.
>
> Since the quantile definition in Line 119 ensures $F_{P_{V|x}}\left(\tau(x,\\{V _i:X _i=x\\} _{i=1} ^n)\right)\geq 1-\alpha$, combining the two inequalities gives
>
> $F _{Q _{V|x}}\left(\tau(x,\\{V _i:X _i=x\\} _{i=1} ^n)\right)\geq 1-\alpha-\text{CCG}(P,Q,x)$,
>
> which is exactly
>
> $\Pr\left(Y _{n+1}\in C _A(X _{n+1})|X _{n+1}=x\right)\geq 1-\alpha-\text{CCG}(P,Q,x)$.
>
> To evaluate the expected conditional coverage across the test distribution, we take the expectation over $x\sim Q _X$
>
> $\mathbb{E}_{x\sim Q_X}[\Pr(Y _{n+1}\in C _A(X _{n+1})|X _{n+1}=x)]\geq 1-\alpha-\text{ICG}(P,Q)$,
>
> where the **Integrated Coverage Gap** $\text{ICG}(P,Q)$ is defined in Eq. (5) as the expectation of $\text{CCG}(P,Q,x)$ over $Q_X$.
>
> Finally, using our bound on $\text{ICG}(P, Q)$ in terms of the Wasserstein distance $W(P_{XY}, Q_{XY})$ in Eq. (14), we obtain a conservative bound on the expected conditional coverage under distribution shift:
>
> $\mathbb{E}_{x\sim Q_X}[\Pr(Y _{n+1}\in C _A(X _{n+1})|X _{n+1}=x)]\geq 1-\alpha-\sqrt{2\kappa L}\left(\eta\cdot W(P _{XY},Q _{XY})+1/4\right)$.
>
> This restatement explicitly connects conditional coverage, our proposed gap metrics, and the extent of distribution shift measured by the Wasserstein distance. We believe this presentation clearly addresses the reviewer’s concern.
>
> We also take this form to update the expected conditional coverage lower bound in **Appendix O.2**, and incorporate the approximation error arising from CQR.

---

### Meta-Review · Area_Chair_NL3d · 2026-01-06

**Summary:**

The paper studies how conformal prediction’s conditional coverage degrades under calibration–test distribution shift and introduces the Conditional Coverage Gap (CCG) and its expectation to quantify this gap, with a Wasserstein-based bound linking coverage degradation to joint distribution shift. Motivated by this theory, it proposes Branched Normalizing Flows (BNF) to transport test data toward the calibration distribution and then predict, yielding improved conditional-coverage robustness in multi-source shift experiments while maintaining marginal coverage.

Across reviewers, the decision was driven by several core concerns. First, reviewers questioned whether the proposed Conditional Coverage Gap (CCG/ICG) truly captures conditional coverage, noting that the paper blends different concepts—such as adaptivity, approximate conditional CP in i.i.d. settings, and distribution shift, without a clear separation.
Second, despite claims of a general distribution shift, the BNF method appears to rely on strong assumptions, access to target features, and settings such as multi-source mixtures. Reviewers have doubted its feasibility and identifiability when target labels are unavailable or when shifts fall outside these structures.
Third, although the Wasserstein-based bound is appealing, reviewers found it hard to interpret and verify in practice, with assumptions and constants that may be loose and difficult to check, limiting its value as a concrete guarantee.
Fourth, while the experiments were broad, concerns remained about the adequacy of slice-based metrics, missing analyses of sensitivity and efficiency–coverage trade-offs, and initially insufficient implementation details.
Finally, presentation issues, unclear definitions, heavy jargon, and limited intuition significantly reduced accessibility.

**Reviewer Concerns:**

Concerns addressed:

1. Link from CCG/ICG to the conditional coverage bound.

2. Requests for missing diagnostics and comparisons, e.g., worst-slice coverage

3. Clarifications around invertibility and feasibility requirements for the mapping $f_\theta$

4. Intuition for why augmentation noise improves expressiveness

5. Missing experimental details

Concerns that remained:

1. Practical informativeness of the Wasserstein bound.

2. Presentation quality and accessibility.

**Reviewer Scores:**

8TcN: 2->4

9Bgi: 4->6

ukCp: may not change.

vg7p: may not change.

---

### Decision · Program_Chairs · 2026-01-26

Reject